



# Reactive nitrogen around the Arabian Peninsula and in the Mediterranean Sea during the 2017 AQABA ship campaign.

Nils Friedrich[1], Philipp Eger[1], Justin Shenolikar[1], Nicolas Sobanski[1], Jan Schuladen[1], Dirk Dienhart[1], Bettina Hottmann[1], Ivan Tadic[1], Horst Fischer[1], Monica Martinez[1], Roland Rohloff[1], Sebastian Tauer[1], Hartwig Harder[1], Eva Y. Pfannerstill[1], Nijing Wang[1], Jonathan Williams[1], James Brooks[2], Frank Drewnick[3], Hang Su[4], Guo Li[5], Yafang Cheng[5], Jos Lelieveld[1], John N. Crowley[1]

[1]Atmospheric Chemistry Department, Max Planck Institute for Chemistry, Mainz, 55118, Germany
[2]Centre for Atmospheric Science, University of Manchester, Manchester, M13 9PL, UK
[3]Particle Chemistry Department, Max Planck Institute for Chemistry, Mainz, 55118, Germany
[4]Multiphase Chemistry Department, Max Planck Institute for Chemistry, Mainz, 55118, Germany
[5]Minerva Research Group, Max Planck Institute for Chemistry, Mainz, 55118, Germany

*Correspondence to*: John N. Crowley (john.crowley@mpic.de)

**Abstract.** We present ship-borne measurements of $NO_x$ ($\equiv NO + NO_2$) and $NO_y$ ($\equiv NO_x$ + gas- and particle-phase organic and inorganic oxides of nitrogen) in summer 2017 as part of the expedition "Air Quality and climate change in the Arabian Basin" (AQABA). The $NO_x$ and $NO_z$ ($\equiv NO_y - NO_x$) measurements, made with a thermal dissociation cavity-ringdown-spectrometer (TD-CRDS), were used to examine the chemical mechanisms involved in the processing of primary $NO_x$ emissions and their influence on the $NO_y$ budget in chemically distinct marine environments, including the Mediterranean Sea, the Red Sea, and the Arabian Gulf which were influenced to varying extents by emissions from shipping and oil and gas production. In all regions, we find that $NO_x$ is strongly connected to ship emissions, both via direct emission of NO and via the formation of HONO and its subsequent photolytic conversion to NO. Mean $NO_2$ lifetimes were 3.9 hours in the Mediterranean Sea, 4.0 hours in the Arabian Gulf and 5.0 hours in the Red Sea area. The cumulative loss of $NO_2$ during the night (reaction with $O_3$) was more important than daytime losses (reaction with OH) over the Arabian Gulf (by a factor 2.8) and over the Red Sea (factor 2.9), whereas over the Mediterranean Sea, where OH levels were high, daytime losses dominated (factor 2.5). Regional ozone production efficiencies (OPE) ranged from $10.5 \pm 0.9$ to $19.1 \pm 1.1$. This metric quantifies the relative strength of photochemical $O_3$ production from $NO_x$, compared to the competing sequestering into $NO_z$ species. The largest values were found over the Arabian Gulf, consistent with high levels of $O_3$ found in that region (10 – 90 percentiles range: 23-108 ppbv). The fractional contribution of individual $NO_z$ species to $NO_y$ exhibited a large regional variability, with $HNO_3$ generally the dominant component (on average 33 % of $NO_y$) with significant contributions from organic nitrates (11 %) and particulate nitrates in the $PM_1$ size range (8 %).





## 1 Introduction

The nitrogen oxides NO and $NO_2$ are emitted into the atmosphere in several natural and anthropogenic processes including lightning (Chameides et al., 1977; Lange et al., 2001), combustion (Lenner, 1987) and bacterial action in soil (Oertel et al., 2016). Due to their rapid interconversion, NO and $NO_2$ are often treated as a single chemical family ($NO_x$). During daytime, NO and $NO_2$ are in photostationary steady-state (R1-R3) (Leighton, 1961), in which the ground-state oxygen atom ($O(^3P)$) generates $O_3$ via the reaction with $O_2$. Ozone can then oxidise NO back to $NO_2$. $O(^3P)$ is produced in the photolysis of $NO_2$

| | | | |
|---|---|---|---|
| $NO_2 + h\nu$ | → | $NO + O(^3P)$ | (R1) |
| $O(^3P) + O_2 + M$ | → | $O_3 + M$ | (R2) |
| $NO + O_3$ | → | $NO_2 + O_2$ | (R3) |

The chemical processing of $NO_x$ in the atmosphere, initiated by ozone and the radicals OH, $HO_2$ and $NO_3$, leads to the formation of $NO_z$ ($NO_z = HNO_3 + NO_3 + 2\,N_2O_5 + RO_2NO_2 + RONO_2 + XONO_2 + XNO_2$ + particulate nitrates) where R is an organic fragment and X represents a halogen atom or an H-atom. The sum of $NO_x$ and $NO_z$ is referred to as total reactive nitrogen $NO_y$ (Logan, 1983), which does not include $N_2$, $N_2O$, $NH_3$ or HCN.

OH, formed e.g. via the photolysis of $O_3$ in the presence of water (reaction R4a and R4b) can directly convert both NO and $NO_2$ to more oxidised, acidic forms (R5, R6):

| | | | |
|---|---|---|---|
| $O_3 + h\nu$ | → | $O(^1D) + O_2$ | (R4a) |
| $O(^1D) + H_2O$ | → | 2 OH | (R4b) |
| $OH + NO + M$ | → | $HONO + M$ | (R5) |
| $OH + NO_2 + M$ | → | $HNO_3 + M$ | (R6) |

or it can react with volatile organic compounds (VOCs) to generate peroxy radicals ($RO_2$, reaction R7). Reaction with organic peroxy radicals converts both NO and $NO_2$ to organic nitrates $RONO_2$ (reaction R9b) or peroxy nitrates $RO_2NO_2$ (reaction R11). In the case of CO, the peroxy radical product is $HO_2$ (R8).

| | | | |
|---|---|---|---|
| $OH + RH\ (+O_2)$ | → | $RO_2 + H_2O$ | (R7) |
| $OH + CO\ (+O_2)$ | → | $HO_2 + CO_2$ | (R8) |
| $RO_2 + NO$ | → | $RO + NO_2$ | (R9a) |
| $RO_2 + NO + M$ | → | $RONO_2 + M$ | (R9b) |
| $HO_2 + NO$ | → | $OH + NO_2$ | (R10) |
| $RO_2 + NO_2 + M$ | → | $RO_2NO_2 + M$ | (R11) |

Reaction with $HO_2$ converts NO to $NO_2$ (R10) and $NO_2$ to $HO_2NO_2$ (R12).

| | | | |
|---|---|---|---|
| $HO_2 + NO_2 + M$ | → | $HO_2NO_2 + M$ | (R12) |

In the mid-latitude lower atmosphere, temperatures are generally sufficiently high that the lifetimes of $HO_2NO_2$ and $RO_2NO_2$ which do not possess an α-carbonyl group (e.g. $CH_3O_2NO_2$) are short (< 1 minute) with respect to re-dissociation to reactants



and they do not represent an important reservoir of $NO_x$. Peroxy nitrates with an $\alpha$-carbonyl group (properly referred to as peroxyacyl acid anhydrides, e.g. PAN) may have lifetimes of a few hours and may temporarily sequester a non-negligible fraction of $NO_x$. The formation of long-lived organic nitrates (R9b) and especially nitric acid (R6) represent daytime sinks for both $NO_x$ and $RO_x$ (OH + $HO_2$ +RO + $RO_2$).

At nighttime, when the photolysis of $NO_2$ ceases, NO is sequentially converted to the $NO_3$ radical (R3, R13) which, via electrophilic addition to unsaturated VOCs in the presence of $O_2$ produces nitrooxy peroxy radicals (reaction R14) which can react with e.g. NO, $HO_2$ or $RO_2$ to form organic nitrates with carbonyl, alcohol and peroxidic substituents (R15) (Ng et al., 2017; Wennberg et al., 2018).

$NO_3$ exists in thermal equilibrium with $NO_2$ and $N_2O_5$ (R16) and the heterogeneous loss of $N_2O_5$ to aqueous surfaces results

in transfer of $NO_y$ to the particle phase as $HNO_3$ (R17) or its loss via deposition. In some (especially marine) environments (Osthoff et al., 2008; Kercher et al., 2009), loss of $N_2O_5$ to particles can result in formation of $ClNO_2$ (R18) which, via photolysis, reforms $NO_2$ the next day.

| | | | |
|---|---|---|---|
| $NO_2 + O_3$ | $\rightarrow$ | $NO_3 + O_2$ | (R13) |
| $NO_3 + R=R \ (+O_2)$ | $\rightarrow$ | $OOR\text{-}RONO_2$ | (R14) |
| 15  $OOR\text{-}RONO_2 + HO_2 / NO$ | $\rightarrow$ | organic nitrate | (R15) |
| $NO_3 + NO_2 + M$ | $\rightleftharpoons$ | $N_2O_5 + M$ | (R16) |
| $N_2O_5 + H_2O$ | $\rightarrow$ | $2 \ HNO_3$ | (R17) |
| $N_2O_5 + Cl^-$ | $\rightarrow$ | $ClNO_2 + NO_3^-$ | (R18) |

The above reactions illustrate that $NO_x$ and VOCs provide the catalyst and fuel for photochemical ozone formation, the

efficiency of which is determined by the competition between photolysis of $NO_2$ to ozone and its conversion to $NO_z$ (Day et al., 2003; Wild et al., 2014; Wild et al., 2016; Womack et al., 2017). Modelling studies have identified the Arabian Gulf as a hotspot for $O_3$ pollution and photochemical smog, with $O_3$ mixing ratios exceeding 100 ppbv (Lelieveld et al., 2009). The lack of measurements in the Arabian Gulf and the Eastern Mediterranean, both of which are expected to be significantly impacted by climate change (Lelieveld et al., 2012), preclude accurate prognosis of air-quality in these regions and provide the rationale

for conducting the AQABA campaign (AQABA: Air Quality and climate change in the Arabian BAsin), in which a large suite of instruments were operated in regions that were influenced by anthropogenic emissions from megacities, petrochemical and shipping activity as well as desert dust emissions and through regions that could be classified as maritime background.

In this paper we present $NO_x$, $NO_y$ and $NO_z$ mixing ratios obtained by a thermal dissociation cavity-ringdown-spectrometer (TD-CRDS), together with a comprehensive set of ancillary measurements and an analysis of the results in terms of

photochemical processing/aging of air masses, and the efficiency of ozone formation.



## 2 Methods

The AQABA ship campaign followed a route from Toulon in Southern France to Kuwait (and back) via the Mediterranean Sea, the Suez Canal, the Red Sea, the Arabian Sea and the Arabian Gulf (see Fig. 1a)). Stops were made in Malta, Jeddah, Djibouti and Fujairah on the first leg (24 June to 30 July 2017), and in Fujairah and Malta on the second leg (2 August to

30 August 2017). Most measurements started in the south-eastern Mediterranean Sea on the first leg and finished ca. half-way between Sicily and Corsica on the second leg. The instruments were located either in air-conditioned research containers aboard, or directly on the deck of the 73 m long research vessel "*Kommandor Iona*". Periods during which instrument inlets were contaminated by ship-stack emissions from the ship (identified based on relative wind direction and speed, and the variability in measured $SO_2$ and NO mixing ratios) were excluded from the analysis. This resulted in rejection of 38.4 % of

the data points on the first leg, when the wind and ship direction were often similar, and rejection of 1.4 % on the second leg of the campaign, when sailing mainly into the wind.

### 2.1 TD-CRDS instrument for $NO_x$, $NO_y$ and $NO_z$ detection

The TD-CRDS instrument, its operating principles and laboratory characterisation, and a validation of the $NO_x$ measurements versus an independent CLD instrument have recently been presented (Friedrich et al., 2020). The TD-CRDS (located in an air-

conditioned research container on the front deck of the vessel) has two separate cavities operating at a wavelength of 405 nm and at sub-ambient pressure (720 to 770 hPa) to prevent condensation of water in inlet lines under humid conditions. Pressure reduction was achieved with a flow restrictor before entering the air-conditioned container. With a sampling flow rate of 3.0 L (STD) min$^{-1}$ (slm) the residence time inside the cavities is ca. 1.2 s. One of the cavities is connected to an inlet (PFA tubing) at ambient temperature, the other cavity is connected, via a 3-way valve, to either of two tubular quartz inlets, both

heated to 850 °C to thermally dissociate $NO_y$ trace-gases to NO or $NO_2$. The TD-ovens were accommodated in an aluminium box on top of the container with the inlets ca. 1.2 m above the container roof. Inlet lines of the heated and the ambient temperature channel were each overall ca. 4 m long (2 m located inside and 2 m outside the container). One of the heated inlets was equipped with an activated carbon denuder in order to remove gas-phase $NO_y$ species from the sample stream and thus, in principal, was used to measure only particulate nitrate. The oven connected to the inlet featuring the denuder

malfunctioned very early in the campaign so that TD-CRDS measurements of particulate nitrate are not available (a short time frame is discussed in Friedrich et al. (2020)). In the Red Sea and Arabian Gulf, the oven heating the inlet of the total-$NO_y$ channel was also switched off occasionally during the hottest hours of the day, to prevent damage to the oven electronics. PTFE filters (diameter 47 mm, pore size 2 μm) located behind the TD-ovens prevented particles from entering the CRDS systems and were exchanged on a daily basis if access to the container was possible. PFA tubes (80 cm long, 1.27 cm i.d.)

were located in the containers directly upstream of both cavities and provided a reaction volume for conversion of NO (either ambient or formed in the TD-ovens) to $NO_2$ by adding 19 ppmv of $O_3$ (generated using a Pen-Ray lamp).



The $NO_z$ mixing ratios obtained using the TD-CRDS were calculated from the difference between $NO_y$ (without denuder) and $NO_x$ measurements and thus contain a contribution from particulate nitrate. Friedrich et al. (2020) have shown that this instrument measures ammonium nitrate quantitatively, but detects only a fraction ($\approx 25$ %) of sodium nitrate ($NaNO_3$) of 200-300 nm diameter as $NO_x$. The inefficient detection of some non-refractory nitrate species (e.g. $NaNO_3$) means that the $NO_y$

mixing ratios presented below are thus (potentially) lower limits. As $NaNO_3$ is usually associated with coarse-mode aerosol (particle diameter > 1 µm) this also implies that the particle-phase nitrate measured by the TD-CRDS is comparable to that measured by an Aerosol Mass Spectrometer (HR-TOF-AMS, see Sect. 2.2). In marine environments, sea salt aerosol can be the dominant aerosol component (Lewis and Schwartz, 2004). We therefore note that the definition of $NO_y$, in this work, is restricted to non-refractory nitrate particles which can be vaporised by the AMS or in the TD inlet of the CRDS. Nitrate

detection by the AMS is further discussed in Sections 3.2.2 and 3.4.

High loadings of coarse mode particles are associated with high wind speeds, which were encountered on the first leg passing the Strait of Bab al-Mandab, through the Arabian Sea and until the Gulf of Oman, and on the second leg in the Arabian Sea and in the Northern Red Sea. The fractional contribution of coarse-mode particles to the overall mass concentration were derived using data from an Optical Particle Counter (OPC) and via the $(PM_{10}-PM_1)/PM_{10}$ ratio. We see from Fig. S1 that the

impact of coarse mode nitrate may have been largest on both legs in the transitional area between Southern Red Sea and Arabian Sea, where OPC $PM_{10}$ mass concentrations exceeded 150 µg m$^{-3}$ and the coarse mode fraction was consistently > ca. 90 %.

The total uncertainty (at 50 % relative humidity and one minute integration time) amounts to 11 % + 10 pptv for $NO_x$ and to 16 % + 14 pptv for $NO_z$ if we disregard the non-quantitative detection of coarse-mode, non-refractory nitrate (see above).

Detection limits during the AQABA campaign were 98 pptv for $NO_x$, 51 pptv for $NO_y$, and 110 pptv for $NO_z$ and are higher than those reported for laboratory operation owing to problems with optical alignment due to the motion of the ship against the sea. The total uncertainty was calculated by combining the measurement uncertainty stemming from systematic errors with the integration time dependent precision value. Systematic errors, which include uncertainties in the effective cross section of $NO_2$ ($\leq 5$ %), the ratio between physical and optical length of the cavity ($\sim 1$ %), the wavelength stability of the laser diode

($\leq 3$ %) and the cavity temperature and pressure ($\sim 0.5$ %) were propagated in quadrature. The error in the humidity correction is calculated as 20 pptv $\times$ RH $\times$ 0.01. $NO_z$ is derived from the difference between the $NO_y$ and $NO_x$ measurements; its total uncertainty was obtained by propagation of uncertainties thereof. Detection limits are defined as the $2\sigma$ standard deviation between consecutive zeroing periods. The campaign data coverage for $NO_y$ is 65 %, considering only time periods when the ship was moving.

## 2.2 Other measurements

Total organic nitrates (ONs) were measured as the sum of peroxy nitrates (PNs, $RO_2NO_2$) and alkyl nitrates (ANs, $RONO_2$) in a five channel, thermal dissociation cavity-ring-down spectrometer (5C-TD-CRDS, Sobanski et al. (2016)). $SO_2$ and $ClNO_2$ were measured with a Chemical Ionisation Quadrupole Mass Spectrometer (CI-QMS) with 15 s time resolution (Eger et al.,



2019a; Eger et al., 2019b). The detection limit for $SO_2$ and $ClNO_2$ were 38 pptv and 12 pptv, the total uncertainties were 20 % ± 23 pptv ($SO_2$) and 30 % ± 6 pptv ($ClNO_2$). Particulate phase nitrate (pNit) and sulphate concentrations in the $PM_1$ size range were obtained by an Aerosol Mass Spectrometer (Aerodyne HR-ToF-AMS; DeCarlo et al. (2006)) with measurement uncertainties of 30 % and 35 %, respectively for the mass concentrations of $NO_3^-$ and $SO_4^{2-}$. Total aerosol mass concentrations

in the $PM_1$ and $PM_{10}$ size ranges were calculated from particle size distributions, detected with an optical particle counter (OPC, Grimm model 1.109; size range: 250 nm to 32 µm) in a 6 s time resolution and with a 35 % uncertainty. Ozone was measured by optical absorption at 253.65 nm in a commercial ozone monitor (2B Technologies Model 202) with total measurement uncertainty of 2 % ± 1 ppbv and a detection limit of 3 ppbv (at 10 s integration time). NO and $NO_2$ were measured with a chemiluminescence detector (CLD 790 SR, ECO Physics, 5 s time resolution) as described in Tadic et al. (2020), with

total measurement uncertainties of 6 % (NO) and 23 % ($NO_2$) and detection limits of 22 pptv for NO and 52 pptv for $NO_2$, both calculated at a time resolution of 5 s and a confidence interval of 2σ. HONO mixing ratios were measured by a long path absorption photometer (LOPAP; Heland et al. (2001)) with a 3-5 pptv detection limit and a measurement uncertainty of 20 %. A spectral radiometer (Metcon GmbH) measured wavelength resolved actinic flux, which was converted to photolysis rate constants ($J$) for $NO_2$, $NO_3$ and HONO using evaluated quantum yields and cross sections (Burkholder et al., 2015). The overall

uncertainty in $J$ is ca. 10 %, based on the calibration accuracy (Bohn et al., 2008). The data is additionally uncorrected for upwelling UV radiation. OH concentrations were obtained from a custom-built laser induced fluorescence instrument (LIF) (Martinez et al., 2010; Regelin et al., 2013)) with an upper limit total uncertainty of 40 %. Total OH reactivity measurements were performed according to the comparative reactivity method (Sinha et al., 2008), with a 5 minute detection limit of 5.4 $s^{-1}$ and a ca. 50 % total uncertainty, as described in Pfannerstill et al. (2019). HCHO was detected by a commercial instrument

(AL4021, AERO-LASER GmbH) according to the Hantzsch method, and had a relative uncertainty of 13 % (Stickler et al., 2006). Multi-pass absorption spectroscopy using a quantum-cascade-laser was used to measure CO mixing ratios with 20 % uncertainty, and a limit of detection of 0.6 ppbv (Li et al., 2013).

## 2.3 Meteorological data

Temperature, wind direction, wind speed and relative humidity were measured by a weather station (Neptune, Sterela), together

with the GPS position and velocity of the ship. Back trajectories were obtained using the HYSPLIT transport and dispersion model (Stein et al., 2015; Rolph et al., 2017). The trajectories were calculated backwards for 48 hours from the GPS location of the ship with a starting height of 100 m AMSL. The back trajectories were limited to 48 hours as this exceeds the lifetimes of both $NO_x$ and $NO_z$ (see later) and is thus sufficient to indicate potential source regions. Back trajectories displayed in graphs are considered to be representative for the prevailing atmospheric flow conditions when passing the respective areas along the

AQABA ship track.





## 3 Results and discussion

In Fig. S2 we show the complete $NO_x$, $NO_y$ and $NO_z$ time series from the campaign, averaged from the 5 s raw data time resolution onto a 5 min grid. Periods of contamination by the ship's own exhaust are indicated by grey background colouring. The regional variation in $NO_x$ and $NO_z$ during the 2nd leg is illustrated in Fig. 1 which also delineates the campaign into the

"Red Sea" (2-16 July 2017 and 17-24 August 2017), the "Arabian Sea" (16-24 July 2017 and 7-17 August 2017; not individually discussed due to limited data coverage), the "Arabian Gulf" (24-31 July 2017 and 3-7 August 2017), and the "Mediterranean Sea" (24-31 August 2017).

Altogether, 4.8 % of the $NO_x$ measurements during AQABA were below the ca. 100 ppt detection limit of the TD-CRDS instrument, indicating only sporadic occurrence of maritime background conditions. Similar observations were made by Tadic

et al. (2020), with only 3.3 % of the $NO_x$ dataset below 50 pptv in the Arabian Sea, the Southern Red Sea, and the Eastern Mediterranean. In comparison, $NO_x$ mixing ratios below 20 pptv were previously found e.g. over the South Atlantic (Fischer et al., 2015). The black lines in Fig. 1a) represent two-day back trajectories (HYSPLIT, see Sect. 2.3). A similar figure for the 1st leg is given in Fig. S3. For the Mediterranean Sea, the Red Sea and the Arabian Gulf we present an analysis of the lifetimes and sources of $NO_x$ and $NO_z$. The chemically distinct regions are compared and contrasted in sections 3.5 and 4.

### 3.1 Mediterranean Sea

Owing to unfavourable winds resulting in contamination of the measurements by the ships own exhaust as well as instrument malfunction, very little useable data was obtained by the TD-CRDS during the first leg through the Mediterranean Sea and we analyse only the data obtained on the return leg (24-31 August 2017). In this period, temperatures varied between 24 °C and 29 °C with relative humidity between 52 % and 89 % (see Fig S2). During most of the transit through the Mediterranean Sea,

winds were from the north. At the end of the cruise when approaching Sicily we encountered a shift in wind direction with air arriving from the north-west. Back trajectories (see Fig. 1a) indicate that when sailing through the Eastern Mediterranean Sea we encountered air masses that had passed over Turkey; the air we sampled in the central Mediterranean Sea had passed over the Balkan states and in the Western Mediterranean it had passed over Greece and Italy. The trajectories ending at the ships location were persistently located in the boundary layer (height < 1000 m) for the previous 48 hours. An exception was the

back trajectory originating from the Black Sea, which was located at a height (above ground level) of up to 1740 m. The back trajectory passing over the island of Crete was located at a maximum height of 3224 m, which may be the result of orographic uplift caused by the central Cretan mountain range.

### 3.1.1 $NO_x$

$NO_x$ mixing ratios were generally low in the Mediterranean Sea (Fig. 2a)). One-minute mean and median mixing ratios of $NO_x$

as detected by the TD-CRDS were 1.3 and 0.3 ppbv, respectively. For the CLD measurements of $NO_x$, the equivalent values are 1.1 ppbv and 0.2 ppbv, respectively. For both instruments, the difference between mean and median values stems from the





frequent occurrence of $NO_x$ plumes resulting from emissions of nearby ships. The $NO_x$ mixing ratios measured by TD-CRDS and CLD were in good agreement (see Friedrich et al. (2020)) and the bias of the TD-CRDS to higher values reflects the exclusion of data below the detection limit. A histogram of the $NO_x$ measurements made by the CLD is displayed in Fig. 2b) which indicates that 33 % of the $NO_x$ data were between 100 and 250 pptv, and 24 % above 1 ppbv. The maximum mixing

ratio of $NO_x$ in the Mediterranean Sea of 84.7 ppbv was measured in the narrowest part of the Strait of Messina, a busy corridor for international shipping with ferry traffic between Italy and Sicily, crossing the *Kommandor Iona*'s ship track. This observation highlights the importance of $NO_x$ shipping emissions in some parts of the Mediterranean Sea, which we return to later.

Potential non-shipping sources of $NO_x$ in this region can be identified via the back trajectories plotted in Fig. 1a): In the eastern

part of the Mediterranean Sea, the air masses were influenced by emissions from the heavily populated and industrialized western Turkish coastal area, the island of Crete and mainland Greece. However, as we show below, the lifetime of $NO_x$ is generally less than 6 hours and the greater fraction of any land-based $NO_x$ emissions would have undergone oxidation to $NO_z$ during the 48 hour transport time of the back trajectory. In the Western Mediterranean Sea, the two-day back trajectories end above the open ocean.

Our data can be compared to results from previous measurements of $NO_x$ in the Mediterranean area. Excluding pollution events, Mallik et al. (2018) report NO and $NO_2$ levels below 0.05 ppbv and 0.25 ppbv, respectively, during the 2014 Cyprus-based CYPHEX campaign in the Eastern Mediterranean Sea. Plume-like increases in $NO_x$ were associated with enhanced $SO_2$ and related to emissions from shipping (Eger et al., 2019b). During the MINOS campaign on the island of Crete, median $NO_2$ mixing ratios between 0.3 ppbv and 0.7 ppbv were reported (Berresheim et al., 2003). The lower mixing ratios were associated

with air masses arriving from the Western European free troposphere, whereas the higher values were air masses impacted by biomass burning in Eastern Europe. In contrast, higher $NO_2$ mixing ratios (typically between 4 ppbv and 6 ppbv excluding plumes) were reported from shipboard measurements in the Aegean Sea (Večeřa et al., 2008). Satellite based observations of $NO_2$ vertical column densities over Crete and in the region between Crete and Sicily, were used to derive near-surface $NO_2$ mixing ratios of up to ~ 0.4 ppbv (Ladstatter-Weissenmayer et al., 2003; Ladstatter-Weissenmayer et al., 2007).

Our $NO_x$ measurements are thus broadly consistent with previous measurements in the Mediterranean Sea which indicate mixing ratios of less than 1 ppbv in the absence of recent emissions from ships. The higher mixing ratios reported by Večeřa et al. (2008) are likely to be related to the close proximity of their ship to $NO_x$ sources on the European continent and denser ship traffic compared to the more southerly AQABA route through the Eastern Mediterranean Sea.

### 3.1.2 $NO_z$

Figures 2c) and d) show a time series and histogram of $NO_z$ for the Mediterranean Sea. The shape of the distribution indicates that $NO_z$ mixing ratios close to the detection limit were rarely measured. The mean (0.8 ppbv), median (0.7 ppbv), maximum (2.8 ppbv) and minimum $NO_z$ mixing ratios (< 0.1 ppbv) along with the narrower distribution indicate that, as expected, $NO_z$ is significantly less variable than $NO_x$. The ratio of the median mixing ratios $NO_z / NO_y$ in the Mediterranean Sea is ~ 0.8;



concomitantly, that of $NO_x / NO_y$ is ~ 0.2. A more detailed analysis of the relative contributions of $NO_x$ and $NO_z$ to $NO_y$ in which we divide the Mediterranean Sea into 7 sub-regions, is presented in the following paragraphs.

The pie charts in Fig. 3, indicate the regional average contributions (in sub-regions, M1 to M7) of reactive nitrogen species to $NO_y$. The fractional contributions are based on measurements of $NO_x$, $NO_y$, gas-phase organic nitrates (ON), particulate nitrate (pNit), $ClNO_2$ and HONO. $HNO_3$ was not measured directly but calculated from $HNO_3 = NO_y - (NO_x + ON + pNit + ClNO_2 + HONO)$, where pNit refers to sub-micron particulate nitrate as measured by the HR-ToF-AMS. Detection of coarse mode pNit by the TD-CRDS (see Friedrich et al. (2020)) would lead to an overestimation of $HNO_3$. However, given that the thermal dissociation to $NO_2$ of $NaNO_3$ particles with 300 nm diameter is inefficient (~ 20 %) with this instrument, a significant bias by coarse mode nitrate (e.g. associated with sea salt or mineral dust) appears unlikely.

In all sub-regions, $HNO_3$ is the dominant component of $NO_z$, with sub-micron pNit only contributing between 5.4 % (M6) and 15.5 % (M2) to $NO_y$, and ONs between 7.1 % (M4) and 16.9 % (M2). $ClNO_2$ only constitutes a minor part of $NO_y$ with ca. 1 % contribution in all regions where $ClNO_2$ was measured (M1-M6). The low mixing ratios of $ClNO_2$ have been attributed to high night-time temperatures and high reactivity of $NO_3$ which reduce the interaction of $N_2O_5$ with chloride containing particles (Eger et al., 2019a). Elevated HONO mixing ratios (up to 0.3 ppbv) were observed in regions M3 and M6 where its contribution to $NO_z$ was 3.8 % and 4.2 %, respectively. As the daytime lifetime of HONO is short (a few minutes) due to its rapid photolysis (Platt et al., 1980), HONO levels up to 0.3 ppbv imply strong sources. Elevated HONO mixing ratios in ship plumes have been observed in previous field measurements (Večeřa et al., 2008; Sun et al., 2020), and could explain the presence of HONO in sub-regions M3 and M6. Other sources of HONO, summarised in Elshorbany et al. (2012) include heterogeneous / photochemical reactions of $NO_x$ and $NO_z$ on various surfaces and also the photolysis of particulate nitrate (Meusel et al., 2018).

Figure 3 also plots the $NO_z / NO_y$ ratio along the ships track. The highest values with median $NO_z / NO_y > 0.68$ were found in regions M2, M4 and M7, reflecting a lack of local $NO_x$ sources as confirmed by the back trajectories. In contrast, the regions designated M3, M5 and M6 are influenced by land-based pollution and are characterised by low $NO_z / NO_y$ ratios (medians < 0.55) reflecting the higher levels of $NO_x$ which contributed 52 % (M3 and M5) and 43 % (M6) to total $NO_y$.

### 3.1.3 Lifetime and sources of $NO_x$

In the following section, the observations of $NO_x$ in the Mediterranean Sea are analysed in terms of its production and loss. Following the considerations in Sect. 1, we compare the daytime loss of $NO_x$ via the reaction between $NO_2$ and OH (R6; expected to dominate over other daytime $NO_x$ loss processes in the marine environment) with night-time losses via the reaction between $NO_2$ and $O_3$ (R13):

$$k^{NO2} = k_6[OH] + k_{13}[O_3] \qquad (1)$$

Where $k^{NO2}$ represents the total loss rate constant (in s$^{-1}$) for $NO_2$ and is the inverse of the $NO_2$ lifetime ($\tau^{NO2}$). The first term on the right-hand-side of this expression is most important at day when OH levels were high (up to $1.4 \times 10^7$ molecules cm$^{-3}$





at local noon) but relatively unimportant at night. In contrast, the second term is only important at night as the $NO_3$ product of R13 is rapidly photolysed back to $NO_x$ during daytime, so that $NO_x$ is conserved. In order to fill gaps in the OH dataset (daytime data coverage of 71 %) complete diel cycles of OH were generated by scaling measurements of $J_{O1D}$ to the OH noon-time maxima. Figure S4 compares the measured OH concentrations with the interpolated trace and shows that the thereby derived

OH levels can be considered as upper limits. Inserting these values and the measured $O_3$ concentration into Eq. 1 and using preferred rate coefficients for $k_6$ and $k_{13}$ (IUPAC, 2020) we derive lifetimes (Fig. 4a)) of ~ 2 hours at local noon (largest OH levels) and 5-6 hours at night. Loss of $NO_x$ by deposition may be important in forested regions (Delaria et al., 2018; Delaria and Cohen, 2020) but is expected to be insignificant in a marine environment. The relative importance of day- and night-time losses of $NO_2$ in the Mediterranean Sea during AQABA was estimated by integrating the two loss terms using the available

$NO_2$, $O_3$ and OH data. Averaged over the 6 days of measurements 3.71 ppbv of $NO_x$ was lost per 12-hour day and 1.51 ppbv per 12-hour night (Fig. 4c)).

Although our conclusion is based on a limited dataset, we calculate that the OH-induced, daytime loss of $NO_x$ is most important in the Mediterranean Sea, reflecting the high levels of OH encountered during AQABA, but note that night-time losses make a significant contribution. It is very likely that in other seasons with reduced photochemical activity and lower temperatures

(which favour the formation of $N_2O_5$ which can remove two $NO_2$ molecules via heterogeneous processes), the night-time losses gain in relative importance. Averaged over the entire dataset obtained in the Mediterranean Sea, we calculate a lifetime of $NO_2$ of 3.9 hours. Chemical sources of $NO_x$ in the Mediterranean Sea, i.e. from the photolysis of HONO and pNit, as well the reaction of OH and $HNO_3$, are discussed in detail in Sect. 3.4.

In the following, we examine the contribution of ship emissions to the $NO_x$ budget in the Mediterranean region and especially

along the track taken by the *Kommandor Iona* during the AQABA campaign. In Fig. S5 we plot a time series of $NO_x$ and $SO_2$ data for the transit through the Mediterranean Sea. It is immediately apparent that large, plume like features in $NO_x$ coincide with similar features in $SO_2$. We now separate the dataset into two regimes in which the $NO_z$ and $NO_y$ measurements indicate either relatively "fresh" emissions ($NO_z / NO_y$ ratio < 0.4) or relatively "aged" emissions ($NO_z / NO_y$ ratio > 0.8). In Fig. 5a) we show that, for fresh emissions, $SO_2$ and $NO_x$ are highly correlated (Pearson's R = 0.84) with a slope of $4 \pm 0.1$ ppbv $NO_x$

per ppbv $SO_2$ and an intercept (at zero $SO_2$) of $-1.9 \pm 0.3$ ppbv. This strongly suggests that fresh $NO_x$ emissions are generally accompanied by $SO_2$ and thus indicates that either ships or power plants, e.g. in coastal locations, are the likely sources of a large fraction of the $NO_x$. The slope is similar to that derived by Celik et al. (2020) ($2.7 \pm 0.8$) who examined single ships plumes in a more detailed analysis and with literature values that range from $6.8 \pm 6.3$ near the coast of Texas (Williams et al., 2009) to $11.2 \pm 10.9$ (Diesch et al., 2013) at the Elbe river near Hamburg/Germany. In comparison to Celik et al. (2020),

however, the two other literature studies only sampled very fresh and unprocessed ship plumes, from a distance of less than ca. 5 kilometres to the emission source. Fig. 5a) shows that $NO_z$ and $SO_2$ are not correlated (Pearson's R = 0.38) in air masses impacted by fresh emissions.

In more aged air masses (Fig. 5b)) the slope of $NO_x$ per $SO_2$ is, as expected, much smaller ($0.16 \pm 0.01$ ppbv $NO_2$ per ppbv $SO_2$) which reflects the significantly longer lifetime of $SO_2$ (~ 10 days) compared to $NO_x$. After a few days of transport an air





mass containing co-emitted $NO_x$ and $SO_2$ will still contain $SO_2$ but the initially emitted $NO_x$ will, to a large extent, have been converted to $NO_z$. The intercept ($NO_x = 0.049 \pm 0.005$ ppbv at zero $SO_2$) is consistent with the re-generation of $NO_x$ from $NO_z$ (see above), but is also in the area of the detection limit of the $NO_x$ measurement.

The plot of $NO_z$ versus $SO_2$ for aged emissions indicates a significant intercept (at zero $SO_2$) of 0.4 ppbv $NO_z$. As the lifetime

of $SO_2$ (~ 10 days) is longer than of $NO_z$ (~ half a day) (Dickerson et al., 1999; Romer et al., 2016) the residual $NO_z$ at zero $SO_2$ cannot stem directly from ship emissions (or combustion sources that generate both $NO_x$ and $SO_2$) but represents the background level of $NO_z$ in the Mediterranean Sea in aged air masses and is consistent with an average $HNO_3$ mixing ratio of 0.48 ppbv observed during the MINOS campaign at Finokalia on Crete (Metzger et al., 2006).

The analysis above, when combined with back trajectory information, provides clear evidence that shipping emissions are

responsible for a large fraction of $NO_x$ in the Mediterranean Sea. The impact of shipping emissions on the atmospheric sulphur budget has been assessed in numerous studies which identify coastal areas and international shipping lanes as important hot spots for $SO_2$ emissions (Capaldo et al., 1999; Dalsoren et al., 2009; Eyring et al., 2010) with emissions of $SO_2$ severely impacting air quality in port-regions (Isakson et al., 2001; Cooper, 2003; Saxe and Larsen, 2004; Marmer and Langmann, 2005; Ledoux et al., 2018). A detailed analysis of $SO_2$ data with regard to ship emissions during AQABA is provided by Celik

et al. (2020) who analysed emission factors from individual ships plumes during the AQABA campaign.

## 3.2 Red Sea

Measurements over the Red Sea (from the Suez Canal and the Strait of Bab al-Mandab) were made from 2-16 July 2017 on the first leg and 17-25 August 2017 on the second leg. On the first leg, the *Kommandor Iona* reversed direction in the Northern Red Sea three times (twice for nine hours and once for six hours), in order to sail into the wind and avoid contamination by

the ship's own stack. Additionally, there was a three-day layover in Jeddah (10 to 13 July 2017). Temperatures on the first leg were usually above 27 °C, with maxima of 37-38 °C in the Suez Canal, in Jeddah and on the approach to Bab al-Mandab. The relative humidity was usually between ca. 60 % and 80 %, but dropped below 30 % in the Suez Canal and in Jeddah. Winds came predominantly from northerly directions with speeds generally between 2 and 10 m s$^{-1}$. On the second leg, temperatures were constantly above 30 °C in the Southern Red Sea, relative humidities were similar to the first leg. The wind was

consistently from the north, with wind speeds between 5 and 12 m s$^{-1}$ until the ship reached the Suez region. During the first leg, the air masses intercepted above the Northern Red Sea were impacted by emissions from Cairo and the Nile valley. Two-day back trajectories for the Southern Red Sea start in the centre of the Red Sea and do not indicate transport from the Suez Region. Extended back trajectories for the Southern Red Sea showed that three to four days prior to sampling, the air parcel passed over southern Egypt, and five to six days before was located over the Cairo area. Similar back trajectories were obtained

for the second leg. Air masses in the Northern Red Sea were influenced by the Suez region, north-eastern Egypt and Israel.



### 3.2.1 NO$_x$

NO$_x$ mixing ratios in the Red Sea (excluding the three day layover in the port of Jeddah) as measured by the TD-CRDS and the CLD instruments are displayed in Fig. 6a). NO$_x$ mixing ratios were highly variable and there were only short periods free of NO$_x$ plumes > 10 ppbv (e.g. during the second leg on 19 and 20 August 2017). The mean NO$_x$ mixing ratios (2.8 ppbv

measured by the TD-CRDS and 3.2 ppbv measured by the CLD) were therefore significantly higher than the median values of 1.0 ppbv. Figure 6b) indicates that the NO$_x$ mixing ratios are broadly distributed around the median of 1.0 ppbv with 21 % of all data points > 3 ppbv. The highest NO$_x$ levels during AQABA were found in narrow shipping corridors of the Suez region and the Strait of Bab al-Mandab. When excluding the Suez and Bab al-Mandab regions, a median NO$_x$ mixing ratio of 0.7 ppbv can be derived for the maritime central part of the Red Sea.

To the best of our knowledge, in situ measurements in the Red Sea area are not available for comparison with our NO$_x$ data. Satellite based modelling studies show that high NO$_2$ column densities above the Red Sea are associated with shipping emissions (Richter et al., 2004; Alahmadi et al., 2019) which is consistent with our observation of a strong correlation between NO$_x$ and SO$_2$ (see below). Johansson et al. (2017) have estimated a NO$_x$ emission rate of 0.70 ton km$^{-2}$ yr$^{-1}$for the Red Sea (including the Suez Region).

**3.2.2 NO$_z$**

The mean mixing ratio of NO$_z$ over the Red Sea was 1.0 ppbv, with a maximum value of 8.0 ppbv measured in the Gulf of Suez on the first leg. NO$_z$ mixing ratios are narrowly distributed (see Fig. 6d)) around a median value of 0.7 ppbv, with 53 % of the measurements between 0.4 and 1.0 ppbv, and 41 % between 1.0 and 4.0 ppbv.

The NO$_z$ / NO$_y$ ratios along the ships track are plotted in Fig. 7: values > 0.6 were mostly observed over the Northern Red Sea

on the first leg, after leaving the Gulf of Suez. On the second leg, the NO$_z$ / NO$_y$ ratio was higher in the Southern Red Sea. NO$_z$ data coverage was limited in the Red Sea on both legs and the NO$_z$ / NO$_y$ ratio was more variable than values found in the Mediterranean Sea and the Arabian Gulf. The high variability in the NO$_z$ / NO$_y$ ratios is caused by the route of the *Kommandor Iona* along the main shipping lane connecting the Suez Canal and the Gulf of Aden and the frequent sampling of plumes from nearby ships. The observed NO$_z$ / NO$_y$ ratios of < 0.6 in the Red Sea highlight the impact of NO$_x$ emissions from

shipping on the reactive nitrogen budget and the air quality in the Red Sea region (as discussed in Sect. 3.2.1).

For the Red Sea, we have defined four sub-regions in which we calculate the contributions of NO$_x$ and various NO$_z$ species to NO$_y$: these are RS1 on the first leg and RS2, RS3 and RS4 on the second leg. Note that RS1 and RS4 are both located in the Northern Red Sea but the measurements (~5 weeks apart) revealed different chemical characteristics, hence the separate treatment.

Due to poor data coverage, mainly of organic nitrates, we were not able to perform this calculation in further sub-regions on the first leg. In all four regions, NO$_x$ was the largest component of NO$_y$ which results from continuous NO$_x$ input from on shore and shipping emissions.



In RS1 we observed the lowest contribution (36.4 %) of $NO_x$ to $NO_y$ and the largest contribution of ONs (23.8 %) to $NO_y$, over the Red Sea. The latter value is the highest found during the entire AQABA campaign and is comparable to the contribution of $HNO_3$ (30.0 %). In roughly co-located RS4, but 5 weeks later, the $NO_x$ contribution was much larger (69.5 %). The divergent median $NO_x$ / $NO_y$ and $NO_z$ / $NO_y$ for sub-regions RS1 and RS4 can be understood when one examines the air

mass back trajectories for the two legs. On the second leg, strong northerly winds transported $NO_x$ from the highly polluted southern end of the Gulf of Suez to RS4, whereas during the first leg, the back trajectory for RS1 passed (with lower wind speeds) mainly over Eastern Egyptian deserts, with emissions from Cairo requiring 36 hours to reach RS1 during which a significant fraction of $NO_x$ was converted to $NO_z$. We expect that the large contribution of ONs in RS1 is a result of the unique chemical environment at the southern end of the Gulf of Suez and in the Northern Red Sea. A large coherent oil field is located

south of the Gulf of Suez and the coast of Eastern Egypt (Alsharhan, 2003) and the numerous facilities for oil extraction result in abundant emissions of VOCs while the proximity to the Gulf of Suez and the narrowing shipping corridor on the approach to Suez provides the $NO_x$ required for formation of organic nitrates (ONs). Meteorological conditions additionally favored a buildup of ONs during our passage through RS1: elevated wind speeds of up to 11 m s$^{-1}$ coincided with temperatures below 30 °C, which slowed down the thermal decomposition of PAN, compared to the ca. 35 °C regime in the Arabian Gulf. Average

PAN mixing ratios, as measured by CIMS, were 190 pptv in this area, which constitutes ca. 20 % of the total ONs signal. On the second leg in RS4, the fractional contribution of ONs was overshadowed by the stronger impact of $NO_x$ pollution from the Suez region (see above).

In RS1 and RS4 the contributions of HONO and $ClNO_2$ to $NO_y$ were minor (≤ 3 %). RS2 and RS3 are both located in the southern half of the Red Sea. For RS3 we observed the highest contribution (15 %) of AMS-measured particulate nitrate to

$NO_y$, and RS3 was characterised in large parts by coarse mode OPC fractions > 85 % (i.e. $(PM_{10}-PM_1)/PM_{10}$; see 18 and 19 August 2017 in Fig. S1). It is reasonable to assume that the coarse-mode particle mass concentrations in this area was due to sea salt, which reacts heterogeneously with $HNO_3$ to form particle-phase nitrates (Mamane and Gottlieb, 1990). Refractory sea salt aerosol particles in the $PM_1$ size range are, however, not expected to be detectable via AMS (Jimenez et al., 2003), or with only very low efficiency (ca. 1 %) (Zorn et al., 2008).

Region RS2 shows a intermediate behaviour, as $NO_z$ / $NO_y$ increases after leaving Bab al-Mandab and transported air only came from the surrounding Southern Red Sea without being influenced by shore side anthropogenic activities. Here, $NO_x$ and $HNO_3$ contribute 57 % and 27 %, respectively. The relatively high $NO_x$ contribution, considering the remote area, can be explained by sampling ships plumes on the departure from Bab el-Mandab, which led to several $NO_x$ peaks above 10 ppbv (see Fig. 6a)). Consequently, background $NO_x$ levels did also not fall below ca. 1.5 ppbv on the night from 17 to

18 August 2017. Overall, the fractional contributions of $NO_x$ were positively biased by short term spikes in $NO_x$ mixing ratios caused by ship plumes, in all Red Sea sub-regions. The use of mean values to assess the fractional contributions of $NO_y$ species in certain sub-regions is thus a caveat of this analysis, as $NO_z$ signals exhibit less variability during pollution events (see Fig. 6c)). Employing the median values, however, would not allow the relative contributions to $NO_y$ to be assessed.



### 3.2.3 Lifetime and sources of $NO_x$

Analogous to Sect. 3.1.3, we now investigate the day- and night-time chemical losses of $NO_2$ in the Red Sea (see Fig. 8). As described previously, we used an interpolated OH data-set based on a scaling factor between the available OH data and $J_{O1D}$. As OH was not measured over the Red Sea on the first leg, our analysis is restricted to the second leg only. Daytime $NO_2$
lifetimes w.r.t. loss by reaction with OH were usually in a range between 2 and 4 hours, with a minimum of 1.7 hours on 21 August 2017, where the noontime OH concentration peaked at $1.1 \times 10^7$ molecules $cm^{-3}$. Night-time $NO_2$ lifetimes (determined by $O_3$ levels) exhibited a larger variability, but were mostly between 5 and 10 hours. The average (day and night) $NO_2$ lifetime in the Red Sea was 5.0 hours.

Over the entire period of measurements in the Red Sea (8 days and 8 nights) we calculate that a cumulative total of 62 ppbv
of $NO_2$ were lost (Fig 8c)). Despite the shorter lifetime of $NO_2$ at noon, the greater integrated loss of $NO_2$ occurred during nighttime (5.7 ppbv per night on average) when continually high $O_3$ levels (median 54 ppbv) were available. At midday, $NO_2$ mixing ratios are reduced due to the shift in the $NO_2 / NO$ ratio caused by the rapid photolysis of $NO_2$ and also because the OH levels are highest then. On average, daytime loss rates were 2.0 ppbv per day.

In order to assess the contribution of shipping on $NO_x$ emissions, we correlated $NO_x$ and $SO_2$ mixing ratios for freshly emitted
($NO_z / NO_y < 0.4$) and chemically more aged ($NO_z / NO_y > 0.8$) air masses. The results are illustrated in Fig. 9 and summarized in Table 1, which reveal a positive correlation (slope of $3.7 \pm 0.1$ and a regression coefficient R of 0.61) between $NO_x$ and $SO_2$ in air masses containing freshly emitted pollutants. Six data points far above 20 ppbv (range 43-128 ppbv $SO_2$) were excluded, as they would bias the linear regression result. Including these data points lowers the slope to $1.26 \pm 0.04$ and the correlation coefficient R to 0.40. The $NO_x / SO_2$ ratio is thus highly variable throughout the Red Sea, potentially reflecting variable
$NO_x / SO_2$ emission ratios of different vessels, using various fuels, as well as the impact (on $NO_x$) of off offshore oil-drilling rigs and shore-side oil refineries. The latter are most important in the Northern Red Sea whereas shipping emissions dominate in the narrow shipping lanes of the Suez Canal.

For chemically aged air masses, the $NO_x / SO_2$ ratio is $0.20 \pm 0.01$ with R = 0.61, the reduction in slope reflecting the shorter lifetime of $NO_x$ compared to $SO_2$. We find however, that in chemically aged air masses, $NO_z$ and $SO_2$ are highly correlated
(Fig. 9b)) with a slope $NO_z / SO_2 = $ of $1.25 \pm 0.04$ and R = 0.85. The intercept (see Fig. 9b)) at an $SO_2$ mixing ratio of zero is $(0.40 \pm 0.03$ ppbv) which can be taken to be the regional $NO_z$ background mixing ratio (i.e. $NO_z$ formed from $NO_x$ which was not emitted from $SO_2$-containing fuels).

### 3.3 Arabian Gulf

Data over the Arabian Gulf (see Fig. 10) were obtained from 24 to 31 July 2017 (first leg) and 31 July to 3 August 2017
(second leg). During the four-day layover in the harbour of Kuwait, the TD-CRDS was not operational. The highest temperatures during the AQABA campaign were found in the Arabian Gulf with daytime temperatures up to 46 °C at Kuwait harbour and 38-39 °C offshore. Nightime temperatures were constantly above 30 °C on both legs. Offshore relative humidities





were between 60 and 90 % during both legs, wind speeds were generally below 6 m s$^{-1}$, and frequently 1-2 m s$^{-1}$. The Arabian Gulf crossing was divided into four sub-regions, A1 and A2 on the first, as well as A3 and A4 on the second leg (see Fig. 11). Air mass back trajectories indicated that air sampled in the Gulf of Oman originated in Oman, the south-eastern Arabian Gulf was influenced by transport from the central Arabian Gulf and Saudi-Arabia. Inside A1, samples were affected by the eastern coast of Saudi Arabia. When approaching Kuwait (area A2), back trajectories pointed towards Iraq. During the second leg, the Northern Arabian Gulf region was dominated by stagnating air masses, mainly containing emissions from local sources and from the direction of Iran. Air from this area was also transported to the central Arabian Gulf, which is covered by sub-region A3. Local sources from inside the shipping lane were dominant when passing the Strait of Hormuz (A4). The Gulf of Oman experienced influx from the remote Arabian Sea, in contrast to the first leg.

**3.3.1 NO$_x$**

Elevated NO$_x$ mixing ratios were detected by both TD-CRDS and CLD throughout the Arabian Gulf (see Fig. 10a). The TD-CRDS measured mean and median NO$_x$ mixing ratios of 3.3 ppbv and 1.6 ppbv, respectively. By comparison, the CLD measured an average of 4.1 ppbv and a median of 1.8 ppbv. The large difference between median and mean reflects the numerous plumes of high NO$_x$ detected by both instruments (Fig. 10a), the deviation of the TD-CRDS and the CLD data is caused by different data coverage as the CLD continued measuring in the most polluted areas close to Fujairah and Kuwait, while the TD-CRDS was switched to zeroing mode, in order to avoid contamination of the inlet lines. When limiting the comparison to periods where both instruments were operating, very similar median values are obtained, with 1.6 ppbv from the TD-CRDS and 1.5 ppbv from the CLD, respectively. A histogram of the NO$_x$ measurements (CLD data only) made in the Arabian Gulf (Fig. 10b) shows a broad distribution, reflecting high variability in the region, with 77 % of the data points falling into a range between 0.4 ppbv and 10 ppbv and a broad maximum at 1-3 ppbv. The highest NO$_x$ daily maxima were observed near Fujairah (up to 34 ppbv on the first and 153 ppbv on the second leg), in the Strait of Hormuz (26 and 30 ppbv), and when approaching/departing Kuwait (43 and 90 ppbv). The locations of these maxima close to the shore or in narrow shipping corridors, and the plume-dominated time series suggest the influence of mostly local pollution sources of NO$_x$, i.e. from ship traffic or from industrial activities in the shore-side areas of the neighbouring cities. NO$_x$ mixing ratios < 0.5 ppbv were found exclusively in the central part of the Arabian Gulf, which is the widest part (least influence from on-shore activity) with the largest spread of the shipping lanes.

The generally very high levels of NO$_x$ in the Arabian Gulf are consistent with results from satellite measurements which have identified high NO$_2$ tropospheric vertical column densities over the Gulf of Oman, the Strait of Hormuz and the south-eastern Arabian Gulf (Beirle et al., 2004). Model studies estimate a NO$_x$ emission rate of 1.13 ton km$^{-2}$ yr$^{-1}$ for the Arabian Gulf (Johansson et al., 2017). With a NO$_x$ lifetime of 4.0 hours (see Sect. 3.3.3) and a boundary layer height of 1 km (Wu et al., 2008), this emission rate translates to a NO$_x$ mixing ratios of 0.3 ppbv. The lower mixing ratio, compared to the median NO$_x$ observed on AQABA (see above), is likely caused by the averaging of the model over the entire Arabian Gulf water surface



area, whereas the *Kommandor Iona* followed common shipping routes with larger $NO_x$ emissions. To the best of our knowledge, there are no in-situ measurements of $NO_x$ over the Arabian Gulf, with which to compare our data.

### 3.3.2 $NO_z$

The Arabian Gulf featured the highest $NO_z$ levels during the AQABA campaign (see Fig. 10c), with mixing ratios from
< 0.1 ppbv up to 6.9 ppbv (mean $2.0 \pm 1.5$ ppbv (standard deviation), median $1.5 \pm 0.7$ ppbv (median absolute deviation)). The histogram of $NO_z$ mixing ratios (Fig. 10d) shows a maximum in the frequency distribution at 1-3 ppbv, with 73 % of all data above 1 ppbv and 15 % above 4 ppbv. Our results thus indicate that the Arabian Gulf is a hotspot for $NO_z$ formation, a result of high levels of the $NO_x$ and VOCs precursors and also $O_3$. The spatial distribution of the $NO_z / NO_y$ ratio for both legs is presented in Fig. 11. On both legs, $NO_z / NO_y$ ratios above 0.8 were found in the central part of the Arabian Gulf, which results
from the processing of $NO_x$ emissions during transport from the shore to the centre of the Arabian Gulf.
We now examine the partitioning of $NO_y$ into its various components in the four sub-regions (A1-A4) defined above for the Arabian Gulf (Fig. 11). On the approach to Kuwait (A2), winds from the north transported fresh $NO_x$ emissions from cities in Kuwait and Iraq to the ship and $NO_x$ accounted for 81 % of $NO_y$. More aged air masses were found in other regions (A1, A3, and A4) with a roughly equal split between $NO_x$ and $HNO_3$ (both 45-50 %) observed in A3 and A4. The major component of
$NO_z$ was $HNO_3$ in all regions, with significant but very variable contribution from organic nitrates, especially in A1 ($13 \pm 16$ %) where the air masses originated from the eastern coast of Saudi-Arabia, which accommodates numerous facilities for oil and gas extraction and processing resulting in high levels of organic trace gases including alkanes, alkenes and aromatics (Bourtsoukidis et al., 2019). Particulate nitrate contributed only minor amounts to $NO_z$ in the Arabian Gulf which reflects the high temperatures and resultant partitioning of nitrate into the gas-phase. Other $NO_z$ species contributed only weakly to the
$NO_z$ as indicated in Fig. 11.

### 3.3.3 Lifetime and sources of $NO_x$

Analogously to Sect. 3.1.3, we also determined $NO_2$ lifetimes and the cumulative loss of $NO_2$ in the Arabian Gulf. The results are presented in Fig. 12. Limited by the availability of OH data, these calculations include only the time period after 29 August 2017 on the first leg. In the same way as in Sect. 3.1.3 we used an interpolated OH data set in the following calculations.
In the Arabian Gulf, daytime $NO_2$ lifetimes (considering loss by OH) were generally between 2 and 4 hours. Night-time lifetimes were in a similar range, but also occasionally exceeded 10 hours, e.g. when leaving the Arabian Gulf towards the Gulf of Oman and the Arabian Sea on the second leg, where $O_3$ mixing ratios fell below 20 ppbv. The average $NO_2$ lifetime was calculated to be 4.0 hours.
Fig. 12c) shows that 50 ppbv $NO_2$ were lost cumulatively throughout the period of measurements over the Arabian Gulf, with
night-time losses (black data points) being more important than daytime losses (red data points). On average 6.0 ppbv $NO_2$ were lost per night, and only 2.1 ppbv per day. Large night-time compared to day-time losses are related to moderate OH levels





in large parts of the Arabian Gulf (see Fig. 12b)). The daytime average OH concentration was $2.4 \times 10^6$ molecules cm$^{-3}$, while on average 73 ppbv $O_3$ was present. The measured OH concentrations were generally low, given the $NO_x$ and $O_3$ levels in the Arabian Gulf, which may have resulted from its reactions with VOCs. With a loss rate constant of 11.6 s$^{-1}$, the Arabian Gulf was the AQABA region with the largest median OH reactivity (Pfannerstill et al., 2019), with 61 % of the total OH reactivity

attributed to various measured VOCs. The daytime losses of $NO_2$ are therefore indirectly limited by the availability of VOCs from e.g. the oil and gas production (see above).

Via analysis of correlation between $SO_2$ and $NO_2$ (Fig. 13 and Table 1) we can assess the influence of shipping emissions on $NO_x$ mixing ratios in the Arabian Gulf. In air masses recently influenced by $NO_x$ emissions ($NO_z / NO_y < 0.4$), $NO_x$ and $SO_2$ are only weakly correlated (slope = $4.1 \pm 0.2$, R = 0.41) indicating that many different $NO_x$ sources (i.e. not only shipping

emissions) contribute. These might include vehicular traffic and industrial activity (e.g. production of nitrogen-based fertilizers (Khan et al., 2016)) in Kuwait City, the Iraqi city of Basra, as well as in Iranian harbours and offshore oil and gas terminals. Considering the limited $NO_x$ lifetime, the land-based emission sources of $NO_x$ gain in importance over plumes from nearby ships, when approaching the coast. In aged air masses, the slope of the $NO_x$ versus $SO_2$ correlation is $0.11 \pm 0.01$ with a large correlation coefficient (R = 0.72). This indicates that in aged air masses, the $NO_x$ levels are linked to $SO_2$ emissions, which is

consistent with the photolysis of HONO being a major source of $NO_x$ in the region. From the intercept ($SO_2$ mixing ratio = zero = $0.0 \pm 0.3$ ppbv) we would expect negligible background levels of $NO_z$. Overall, shipping was an important source of $NO_x$ in the Arabian Gulf, both through direct emissions and via photolysis of ship-related HONO.

### 3.4 $NO_x$ and $NO_y$ and the role of ship emission-related HONO formation during AQABA

In this section, we perform a steady-state analysis, assessing to what extent chemical source strengths can explain the

background mixing ratios of $NO_x$ observed during AQABA. The required $NO_x$ source strength ($P$, in molec cm$^{-3}$ s$^{-1}$) to maintain the observed $NO_x$ levels, is derived from the measured mixing ratios [$NO_x$], and the $NO_2$ reactivity ($k^{NO2}$; see Sect. 3.1.3), whereby $P = P_{chem} + E$ is a combination of chemical production ($P_{chem}$) and direct emission ($E$). Notably, we neglect direct emissions under background conditions (i.e. $E = 0$), and assume that $NO_x$ is only lost via the reaction of $NO_2$ with OH (i.e. $k^{NOx} = k^{NO2}$).

$$P = [NO_x] \cdot k^{NO2} \tag{2}$$

Chemical processes that result in the formation of $NO_x$ include the degradation of two gas-phase $NO_z$ components, HONO and $HNO_3$ and the photolysis of particulate nitrate.

| HONO + $h\nu$ | $\rightarrow$ | NO + OH | (R19) |
| OH + $HNO_3$ | $\rightarrow$ | $NO_3$ + $H_2O$ | (R20) |
| pNit + $h\nu$ | $\rightarrow$ | 2 HONO + 1 $NO_x$ | (R21) |

In a first step, we examine whether the HONO levels observed on AQABA can be explained by the photolysis of pNit in the PM$_1$ size range. This calculation is based on the assumption of a steady state for HONO established at noon through its



photolytic loss, and its production through the photolysis of pNit. Using average noontime Mediterranean Sea concentrations for HONO ($2.44 \times 10^9$ molec cm$^{-3}$), and pNit ($2.93 \times 10^9$ molec cm$^{-3}$), and a photolysis rate $J_{HONO}$ ($1.45 \times 10^{-3}$ s$^{-1}$), we calculate that a value of $J_{pNit} \sim 1.21 \times 10^{-3}$ s$^{-1}$ would be required in order to maintain the observed HONO concentrations. This is a factor $\sim$ 5-6 higher than a reported value of $J_{pNit} \sim 2 \times 10^{-4}$ s$^{-1}$, based on observations over the western North Atlantic Ocean (Ye et

al., 2016). It is however unclear whether the type and age of particles examined by (Ye et al., 2016) are comparable to those in AQABA. In addition, photolysable nitrate associated with particles that are > 1 µm diameter remain undetected by the AMS and could also contribute to the discrepancy between required and literature $J_{pNit}$.

Laboratory studies have demonstrated the conversion of $NO_2$ to HONO on BC particles, with a clear enhancement under UV irradiation (Acker et al., 2006; Elshorbany et al., 2009; Monge et al., 2010; Ma et al., 2013). Monge et al. (2010) postulated

the transport of HONO and NO to remote low-$NO_x$ areas, enabled via this heterogeneous mechanism. Besides the effect of irradiation, heterogeneous, BC-assisted HONO and NO generation also shows a remarkable humidity dependence (Lammel and Perner, 1988; Kalberer et al., 1999; Kleffmann et al., 1999). Further information on the particulate phase chemistry of HONO can be found in comprehensive reviews by Ma et al. (2013) and George et al. (2015). Sources of HONO during the AQABA campaign will be discussed in more detail, in a separate publication.

Using Eq. 2, we now calculate what values of $P_{chem}$ are required to maintain the background levels of $NO_x$ observed and assess the individual contributions from reactions R19-R21 (results presented in Table 2). The analysis was restricted to data points where $NO_z / NO_y$ was greater than 0.6 and to the four-hour timeframe around local noon, in order to focus on aged air mass conditions during maximum photochemical activity. $NO_2$ reacting with $O_3$ was not considered as a $NO_x$ loss mechanism, due to the rapid reformation of $NO_x$ by the photolysis of $NO_3$. For the pNit photolysis rate constant to form $NO_x$ we used $0.33 \cdot J_{pNit}$

(from (Ye et al., 2016)), which accounts for the HONO / $NO_x$ production ratio of 2:1. Additionally, we scaled $J_{pNit}$ with $J_{HONO}$ (normed to the average daytime maximum of $J_{HONO}$), to introduce diurnal variability. Due to limited data availability and rare occurrence of $NO_z / NO_y > 0.6$ (i.e. sampling of aged air) in the other regions, we performed this calculation for the Mediterranean Sea only. The results indicate that the measured HONO concentrations should result in a factor ca. 4.7 times larger $NO_x$ production term than calculated via Eq. 2. Possible explanations for this include a positively biased HONO

measurement, or the underestimation of $NO_x$ losses, e.g. due to undetected OH (despite the upper limit chosen in the interpolation). Our measurements and calculations, nonetheless, allow the qualitative identification of HONO photolysis as a major source of daytime background $NO_x$ levels during AQABA. The production rate from pNit photolysis can also account for ca. 64 % of the chemical $NO_x$ generation, whereas the reaction of OH and $HNO_3$ forms an order of magnitude less $NO_x$.

Throughout AQABA, shipping emissions were responsible for fresh input of pollutant $NO_x$ into the atmosphere. Our

observation that levels of $NO_x$ (with a lifetime of a few hours) were correlated with $SO_2$ (with lifetimes of more than a week) levels even in aged air masses and the observation that HONO photolysis was an important source of NO may be reconciled by considering that HONO (and thus $NO_x$) production is driven by heterogeneous photochemistry on nitrate containing particulate matter, the formation of which is associated with emissions of $NO_x$ and $SO_2$ as well as black carbon. The latter has a lifetime in the boundary layer (defined by its deposition) of about a week or longer in the absence of precipitation and is thus





comparable to that of $SO_2$. The slow, photochemically induced conversion of nitrate to HONO thus provides a long-lived source of $NO_x$ and a link with $SO_2$, together with an explanation for the detection of short-lived HONO even in processed air masses in the Eastern Mediterranean Sea.

## 3.5 Inter-regional ozone production efficiency (OPE)

The ozone production efficiency quantifies the fractional transformation of primarily emitted $NO_x$ to $O_3$ (R1 to R3, (Liu et al., 1987; Trainer et al., 1993)) and thus reflects the relative importance of competing photochemical processes leading to $O_3$ and $NO_z$ formation from $NO_x$. High values of OPE are favoured by low OH and VOC concentrations and values exceeding 80 have been reported for remote marine environments. Low single digit values have been observed in polluted urban environments (Rickard et al., 2002; Wang et al., 2018).

The OPE can be calculated from the relationship between $O_x$ and $NO_z$ where $O_x = O_3 + NO_2$ and the $O_3$ mixing ratios are augmented by those of $NO_2$, 95 % of which potentially photolyses to $O_3$ (Wood et al., 2009). Note that in any air mass where $HNO_3$ is a major component of $NO_z$, the derived OPE may represent an upper limit if $HNO_3$ is lost during transport from the $NO_x$ source region to the measurement location. The $NO_y / CO$ ratio has been used to estimate the impact of $NO_z$ losses on the values of OPE obtained in this type of analysis (Nunnermacker et al., 2000) the rationale being that CO (like $O_3$) is a product

of photochemical activity and relatively long lived, at least compared to $NO_z$. The high variability in the $NO_y / CO$ ratio during AQABA is however indicative of local (non-photochemical) sources of CO e.g. via combustion and precludes use of this corrective procedure so that the values of OPE we present should be regarded as upper limits.

In Fig. 14 we plot $O_x$ versus $NO_z$ for which the $NO_2$ photolysis rate constant was $> 1 \times 10^{-3}$ s$^{-1}$, which restricts the analysis to hours of the day with active photochemistry. Regional OPE values are $10.5 \pm 0.9$ for the Red Sea, $19.1 \pm 1.1$ for the Arabian

Gulf, and $15.4 \pm 2.4$ for the Eastern Mediterranean Sea. The heterogeneity of $NO_z$ and $O_3$ mixing ratios, i.e. the chemical conditions frequently varying between aged and plume situations (see Sect. 3.1), resulted in a low correlation coefficient in the Western Mediterranean Sea ($R^2 = 0.19$), which precluded derivation of an OPE for this region and led us to restrict the Mediterranean Sea OPE analysis to the more homogenous eastern part (encompassing sub-regions M1-M5).

The range of OPE values measured during AQABA (10.5-19.1) is comparable to the value of 10, derived in the MBL at Oki

Island, Japan, a site which is influenced by pollution arriving from the Korean peninsula and the Japanese mainland (Jaffe et al., 1996), but much lower than the value of 87 which was derived from observations off the coast of Newfoundland (Wang et al., 1996), where the median $NO_x$ mixing ratio was $< 100$ pptv. As alluded to above, high values in remote locations may in part be a result of reactive nitrogen loss via deposition. By comparison, during AQABA the median $NO_x$ mixing ratio was $> 600$ pptv, which together with the relatively low OPE indicates that the vast majority of the AQABA ship track cannot be

considered representative of remote MBL conditions.

Figure 14 and Table 3 indicate that the Arabian Gulf, for which the highest $O_3$ levels in the entire campaign were found (up to 150 ppbv) also has the largest OPE, despite high median $NO_x$ mixing ratios. The high OPE value, however, consistent with the analysis of Pfannerstill et al. (2019) who used VOC and OH reactivity measurements to derive the fraction of OH that



reacts with VOCs (fuelling the formation of $RO_2$, conversion of NO to $NO_2$ and thus $O_3$ formation) versus the fraction that reacted with $NO_x$ (resulting in $NO_z$ formation) to identify regions where $O_3$ formation was $NO_x$-limited, VOC-limited or (as was generally the case) in a transition regime. Pfannerstill et al. (2019) indicated that formation of $O_3$ was favoured around the Arabian Peninsula where VOCs from petroleum-extraction and processing industries were important sinks of OH. The highest

net production rates of $O_3$ (NOPR) during AQABA were also found in the Arabian Gulf where calculations of the rate of $RO_2$ induced oxidation of NO to $NO_2$ resulted in a median (over the diel cycle) value of NOPR = 32 ppbv per day which was driven by high noon-time mixing ratios of $RO_2$ (73 pptv in the Arabian Gulf) (Tadic et al., 2020). In the other two regions, the correlation coefficients are notably smaller, due to the lower span in $O_3$ and $NO_z$, resulting in increased relative errors for the derived OPE values.

In Fig. 15 we plot a time series of $NO_z$ mixing ratios during the transition from the Arabian Sea to the Arabian Gulf along with $NO_2$ photolysis rates, $O_3$ and formaldehyde (HCHO) which is formed during the photochemical processing of many VOCs (Fischer et al., 2003; Klippel et al., 2011; Wolfe et al., 2016; Wolfe et al., 2019) and which can therefore be used as a tracer for photochemical activity (Dodge, 1990; Altshuller, 1993; Garcia et al., 2006; Duncan et al., 2010; Parrish et al., 2012). The transition from low $NO_z$ levels in the Arabian Sea to values up to ~7 ppbv in the Strait of Hormuz (SH) is accompanied by

increases in both $O_3$ (up to 160 ppbv) and HCHO (up to 12.5 ppbv). Based on the analysis by Duncan et al. (2010), Tadic et al. (2020) calculated a median HCHO / $NO_2$ ratio of 9.3 for the Arabian Gulf, indicating that $O_3$ production in this region is $NO_x$-limited. The high levels of $NO_z$, $O_3$ and HCHO in the Arabian Gulf result from the combination of intense solar radiation with high levels of reactive VOCs (Bourtsoukidis et al., 2019; Pfannerstill et al., 2019) and $NO_x$ and are accompanied by the highest levels of gas-phase organic nitrates observed during AQABA, with absolute mixing ratios up to 2.5 ppbv on the

approach to Kuwait. In conclusion, our $NO_x/NO_y$ measurements and the OPE values derived from them confirm the exceptional photochemical activity in the Arabian Gulf.

## 4 Conclusions

During the AQABA campaign in the summer of 2017, we collected a unique $NO_x$ and $NO_y$ dataset that covers the Mediterranean Sea, the Red Sea, and the Arabian Gulf, regions with only few previously published observational data sets.

The highest median $NO_x$ and $NO_z$ mixing ratios were observed in the Arabian Gulf ($NO_x$: 1.6 ppbv; $NO_z$: 1.5 ppbv), followed by the Red Sea ($NO_x$: 1.0 ppbv; $NO_z$: 0.7 ppbv) and the Mediterranean Sea ($NO_x$: 0.3 ppbv; $NO_z$: 0.7 ppbv). Night-time losses of $NO_2$ exceeded daytime losses by factors of 2.8 and 2.9 in the Arabian Gulf and the Red Sea, respectively, whereas daytime losses were 2.5 times higher in the Mediterranean Sea, a result of consistently high daytime OH-concentrations.

The derivation of $NO_x$ lifetimes enabled us to calculate the $NO_x$ source strength required to reproduce the observed mixing

ratios and indicated that HONO photolysis was a significant source of $NO_x$ in the Mediterranean Sea. The strong correlation between $NO_x$ and $SO_2$ in air masses that were impacted by fresh emissions of $NO_x$ indicated that ships are the dominant source

of $NO_x$ throughout the AQABA campaign. HONO may have been generated on particulate nitrate, possibly associated with black carbon that has been processed (to contain sulphate, organics and nitrate) as the ships plumes evolve chemically.

The fractional contributions to $NO_x$ of $NO_y$ and the various components of $NO_z$ were highly variable in the three regions. The lowest regional mean contribution of $NO_x$ to $NO_y$ (i.e. most aged air masses) was found in the Mediterranean Sea with 41 %

compared to 47 % in the Red Sea and 46 % in the Arabian Gulf. Of the $NO_z$ trace-gases, $HNO_3$ represented the most important contribution to $NO_y$ with 39 % in the Arabian Gulf, 25 % in the Red Sea and 35 % in the Mediterranean Sea. A clear regional variability was observed for the contribution of organic nitrates, with the highest value (16 % in the Red Sea) related to the concurrent availability of precursor $NO_x$ and VOCs from the oil and gas industry. Comparable figures were derived for the Arabian Gulf and the Mediterranean Sea, with 10 % and 11 %, respectively. pNit (dp < 1 µm) contributed only a few percent,

with the largest value (10 %) found in the Mediterranean Sea. HONO and $ClNO_2$ were generally only minor components (< 3 %) of $NO_z$. Future studies on the reactive nitrogen budget in the AQABA region might benefit from longer stationary measurements (e.g. to identify diurnal patterns), together with the detection of more speciated $NO_z$ compounds (especially $HNO_3$ and ONs).

**Data availability**

All AQABA data sets used in this study are permanently stored in an archive on the KEEPER service of the Max Planck Digital Library (https://keeper.mpdl.mpg.de; last access: 4 December 2020), and are available to all scientists, which agree to the AQABA data protocol.

**Author contributions**

NF analysed the $NO_x$ and $NO_y$ data sets and wrote the manuscript. NF and JNC operated the TD-CRDS. PE and JNC provided

CIMS measurements of $SO_2$ and $ClNO_2$. JSh, NS and JNC performed and evaluated ONs measurements. JSc set up and processed data from the spectral radiometer. DD, BH, IT and HF contributed NO, $NO_2$, HCHO and CO measurements. MM, RR, ST and HH provided OH concentrations. EYP, NW and JW were responsible for the OH reactivity measurements. JB and FD performed measurements with the AMS and OPC instruments. HS, GL and YC contributed the HONO data set. JL designed the AQABA campaign. All authors contributed to the writing of the manuscript.

**Competing interests**

The authors declare that they have no conflict of interest.





**Acknowledgements**

The authors gratefully acknowledge the NOAA Air Resources Laboratory (ARL) for the provision of the HYSPLIT transport and dispersion model and READY website (https://www.ready.noaa.gov) used in this publication. We thank the whole crew of the *Kommandor Iona* and Hays Ships for their support, as well as Marcel Dorf for organising the campaign.



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





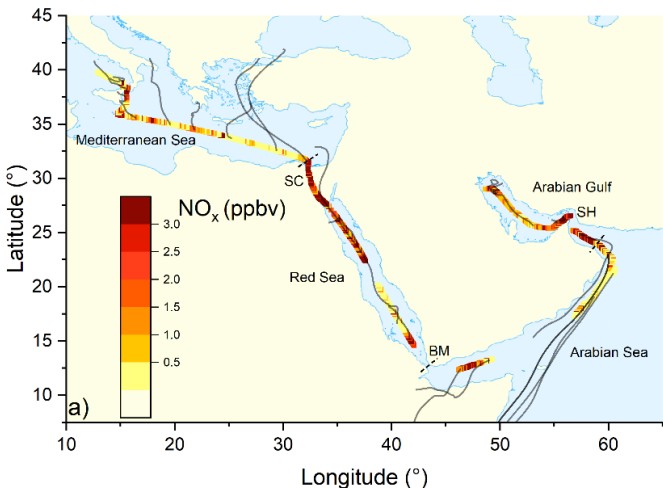

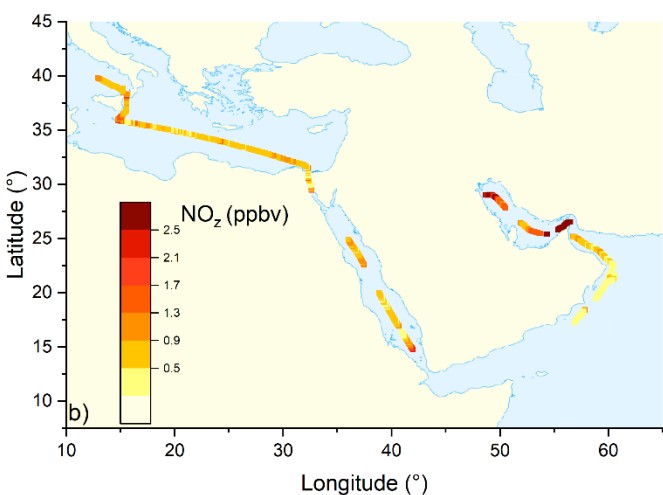

**Figure 1:** Mixing ratios of *(a)* $NO_x$ and *(b)* $NO_z$ from the second leg of the campaign, colour-coded along the ship track. Each data point represents an average over 30 minutes. Grey lines in *(a)* represent HYSPLIT 48 hours back trajectories starting from the ship location at 100 m height. SH = Strait of Hormuz; BM = Strait of Bab al-Mandab; SC = Suez Canal.





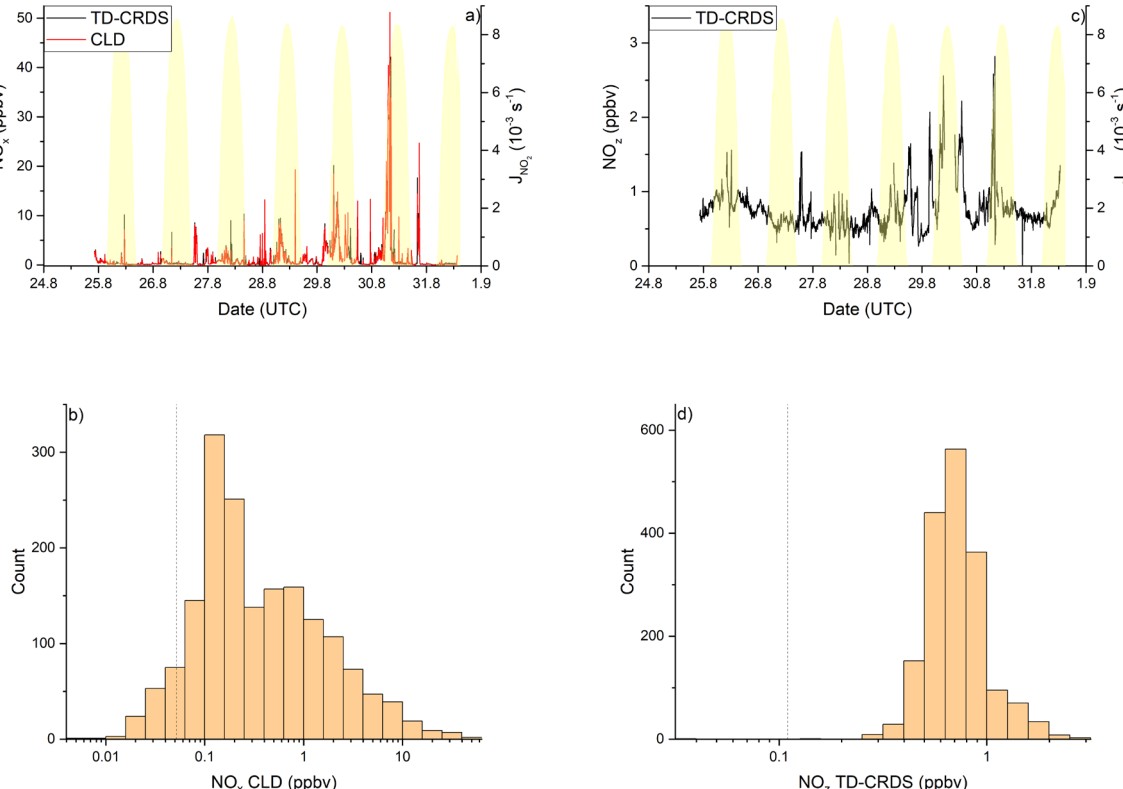

**Figure 2:** $NO_y$ measurements in the *Mediterranean Sea*. Dashed lines signify the instrument detection limits. *(a)* $NO_x$ mixing ratios by CLD and TD-CRDS. *(b)* Frequency distribution of $NO_x$ mixing ratios between 25 August and 1 September 2017. *(c)* $NO_z$ mixing ratios by TD-CRDS. *(d)* Frequency distribution of $NO_z$ mixing ratios between 25 August and 1 September 2017. The yellow shaded regions show $J_{NO2}$. The vertical dotted lines are the limits of detection of the respective measurements.





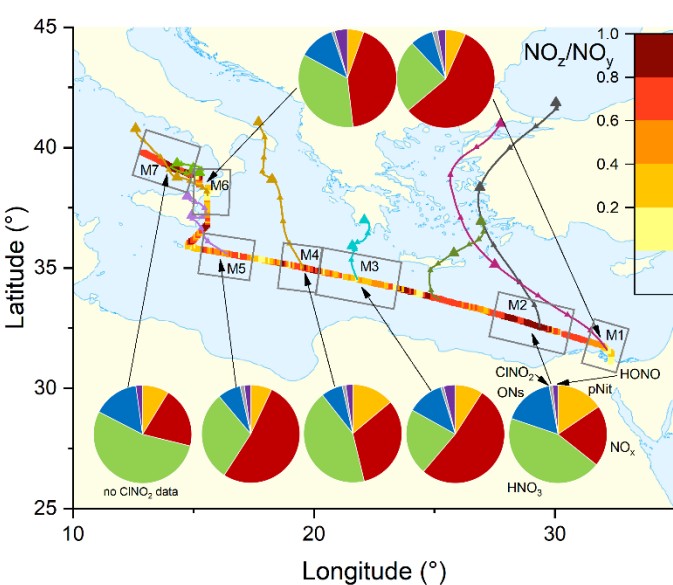

**Figure 3:** The $NO_z$ / $NO_y$ ratio over the *Mediterranean Sea*. Coloured lines are 2-day back-trajectories (HYSPLIT). The pie charts indicate the components of $NO_y$ at various segments along the ship's track (ONs = organic nitrates, pNit = particulate

5    nitrate). $HNO_3$ was calculated via: $HNO_3 = NO_z - (ONs + pNit + NO_x + ClNO_2 + HONO)$.





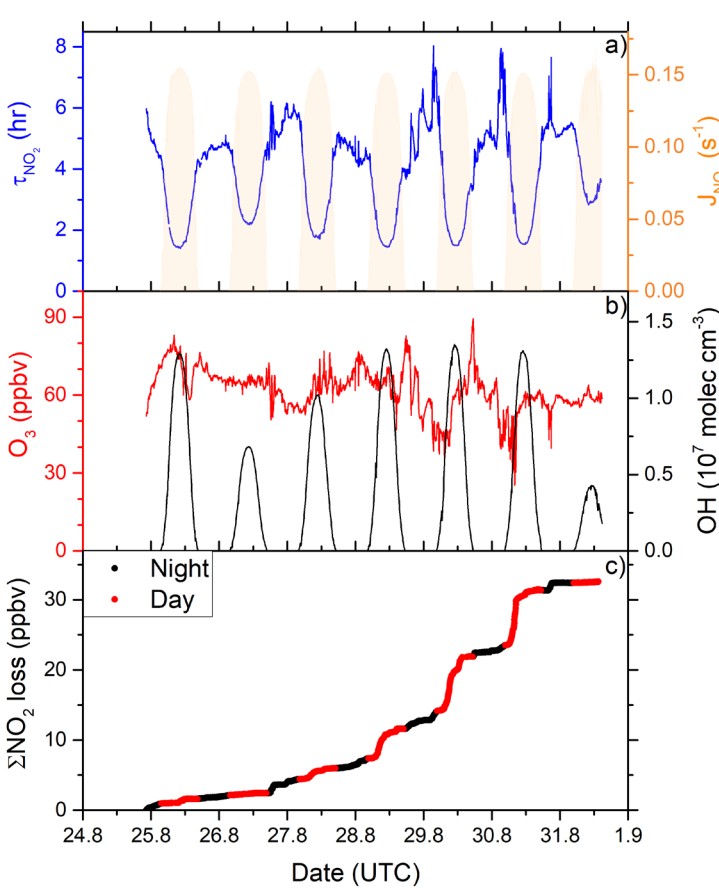

**Figure 4:** *(a)* Lifetime (τ) of $NO_2$ due to reactions with OH and $O_3$ in the *Mediterranean Sea*, together with concentrations of $O_3$ and OH. The OH trace is an interpolation based on OH measurements and $J_{O1D}$ (see Sect. 3.1.3). Daytime hours are indicated via $J_{NO3}$. *(b)* Cumulative loss of $NO_2$ during the displayed time frame, based on the calculated lifetimes and measured $NO_2$.



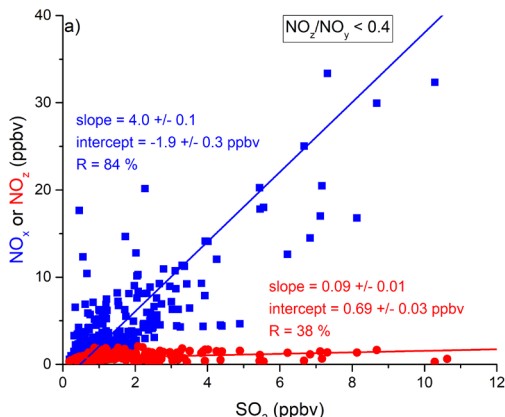

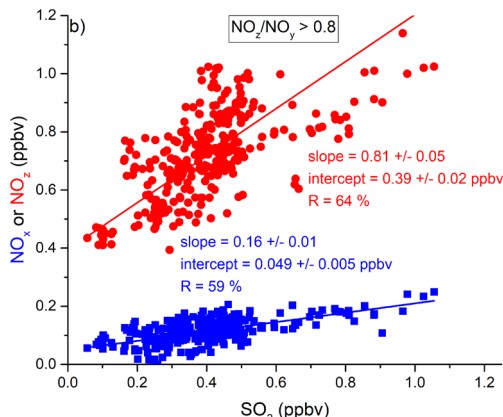

**Figure 5:** Correlation between $SO_2$ and $NO_x$ or $NO_z$ for *(a)* fresh and *(b)* aged $NO_x$ emissions in the *Mediterranean Sea*.





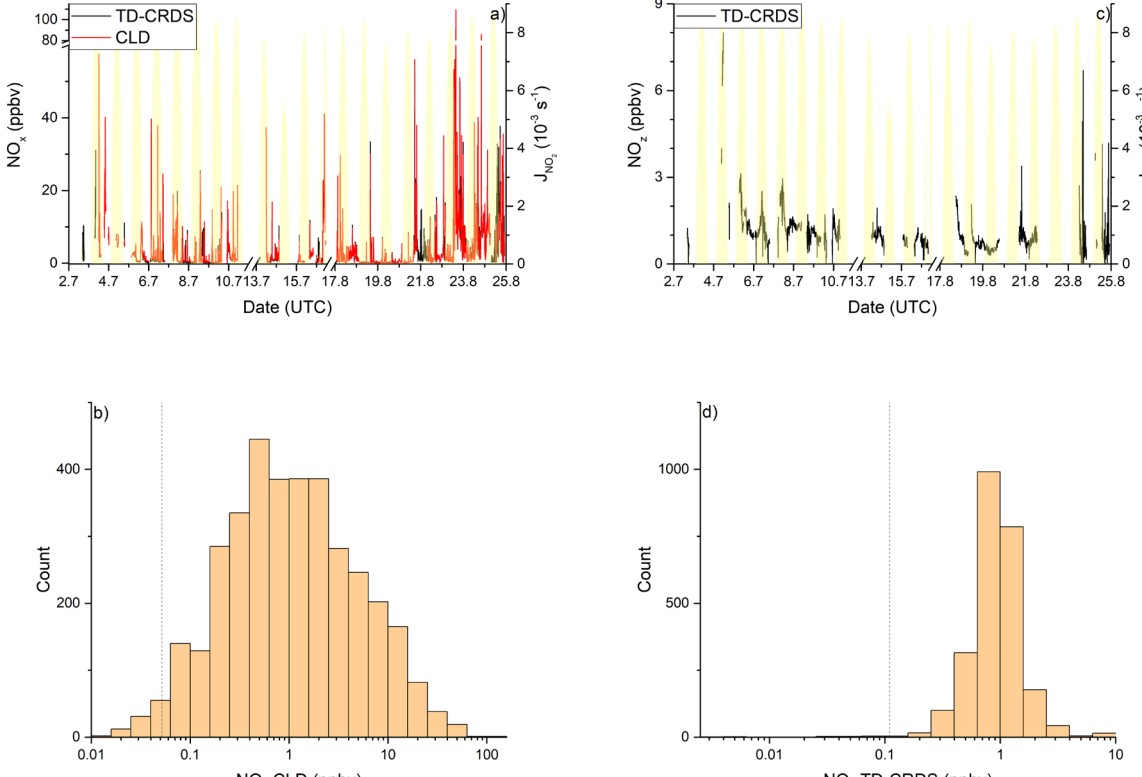

**Figure 6:** $NO_y$ measurements in the *Red Sea*. Dashed lines signify the instrument detection limits. *(a)* $NO_x$ mixing ratios by CLD and TD-CRDS. *(b)* Frequency distribution of $NO_x$ mixing ratios during 2.-16.7.2017 and 17.-24.8.2017, excluding the layover in Jeddah. *(c)* $NO_z$ mixing ratios by TD-CRDS. *(d)* Frequency of $NO_z$ mixing ratios during 2.-16.7.2017 and 17.-24.8.2017. The yellow shaded regions show $J_{NO2}$. The vertical dotted lines are the limits of detection of the respective measurements.





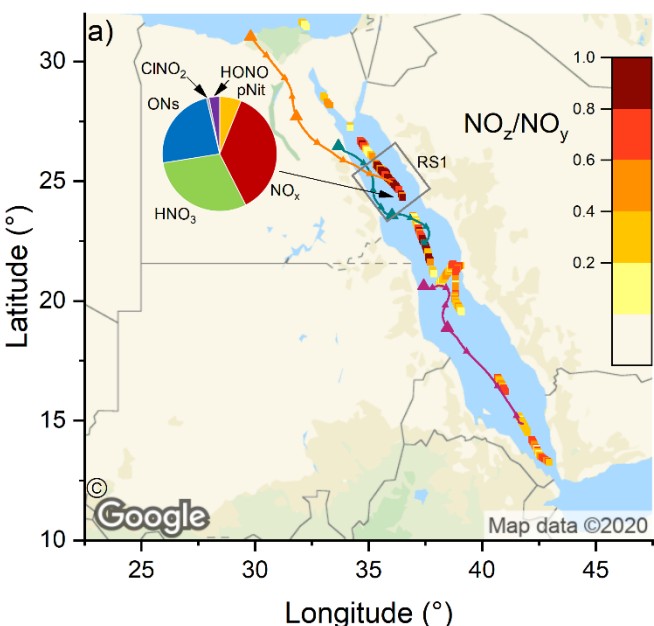

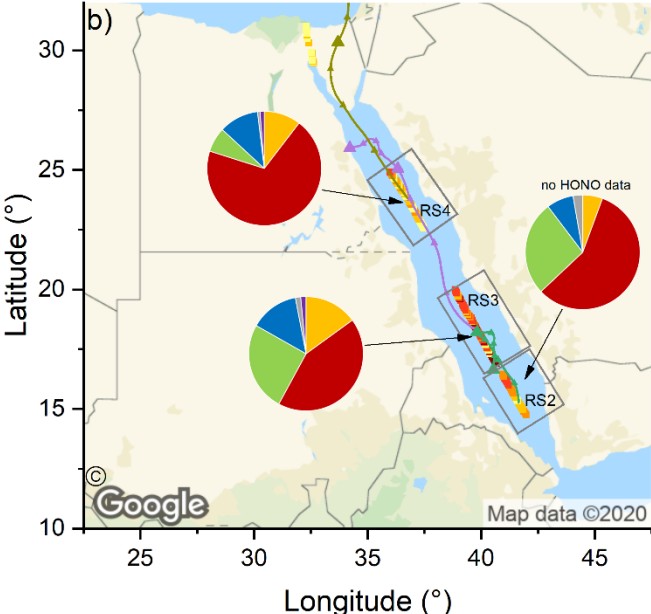

**Figure 7:** The $NO_z/NO_y$ ratio over the Red Sea during the *(a)* first and *(b)* second leg. Coloured lines are 2-day back-trajectories (HYSPLIT). The pie charts indicate the components of $NO_y$ at various segments along the ship's track (ONs = organic nitrates, pNit = particulate nitrate). $HNO_3$ was calculated via: $HNO_3 = NO_z - (ONs + pNit + NO_x + ClNO_2 + HONO)$.


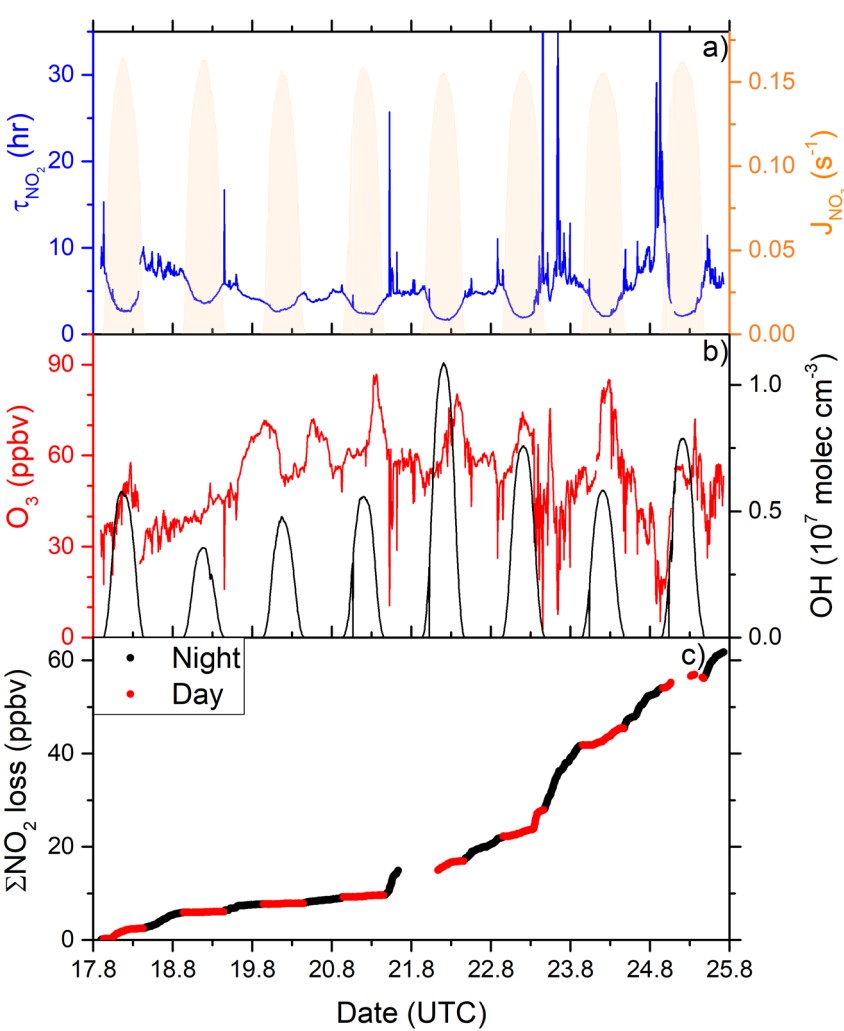

**Figure 8:** *(a)* Lifetime ($\tau$) of $NO_2$ due to reactions with OH and $O_3$ along the second *Red Sea* leg, together with concentrations of $O_3$ and OH. The OH trace is an interpolation based on OH measurements and $J_{O1D}$ (see Sect. 3.2.3). Daytime hours are indicated via $J_{NO3}$. *(b)* Cumulative loss of $NO_2$ during the displayed time frame, based on the calculated lifetimes and measured $NO_2$.



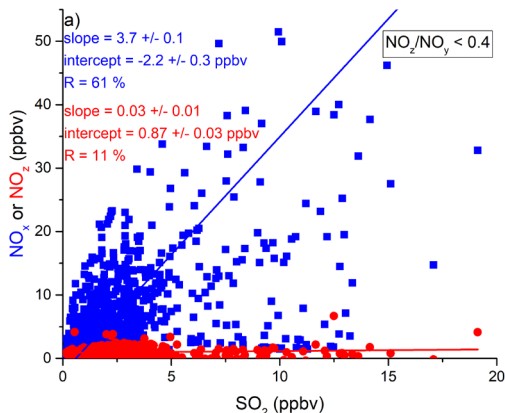

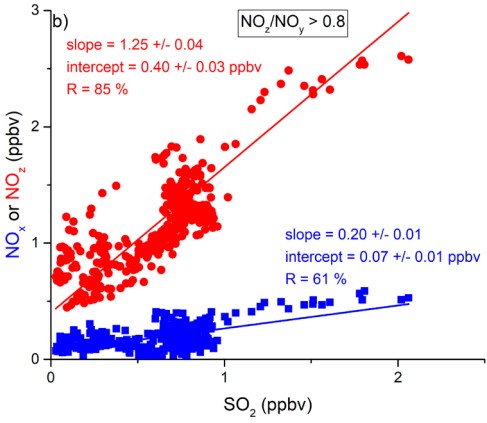

**Figure 9:** Correlation between $SO_2$ and $NO_x$ or $NO_z$ for *(a)* fresh and *(b)* aged $NO_x$ emissions in the *Red Sea*.




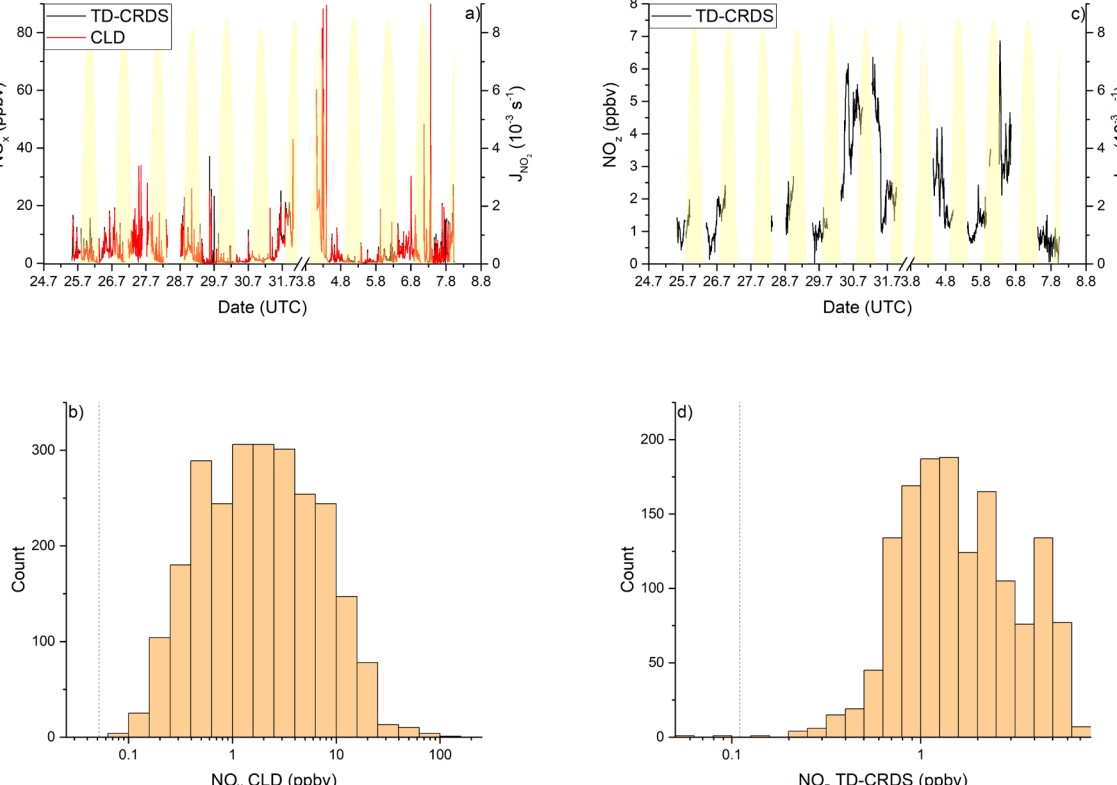

**Figure 10:** $NO_y$ measurements in the *Arabian Gulf*. Dashed lines signify the instrument detection limits. *(a)* $NO_x$ mixing ratios by CLD and TD-CRDS. The $NO_x$ peak in the afternoon of 6.8.2017 reached 153 ppbv. *(b)* Frequency of $NO_x$ mixing ratios between 24.7. and 7.8.2017, excluding the layover in Kuwait. *(c)* $NO_z$ mixing ratios by TD-CRDS. *(d)* Frequency of $NO_z$ mixing ratios between 24.7. and 7.8.2020. The yellow shaded regions show $J_{NO2}$.



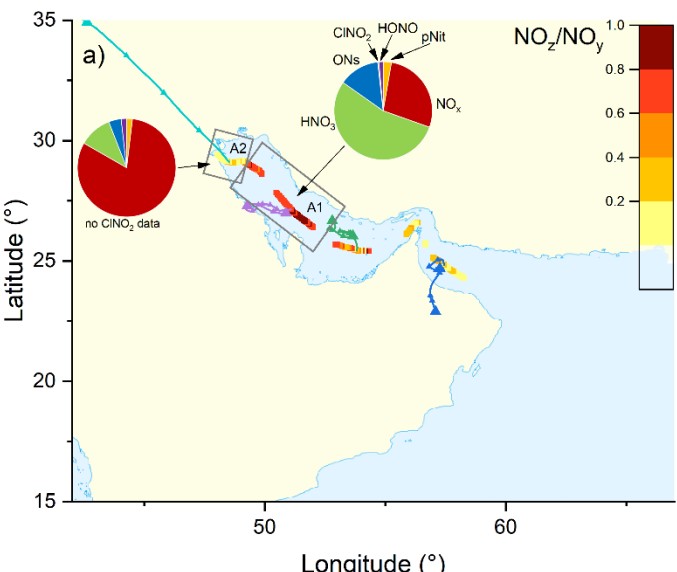

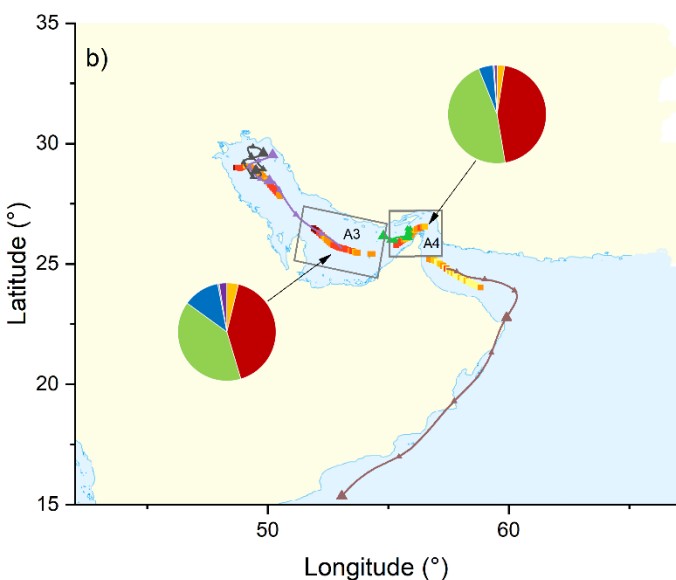

**Figure 11:** The $NO_z / NO_y$ ratio over the Arabian Gulf during the *(a)* first and *(b)* second leg. Coloured lines are 2-day back-trajectories (HYSPLIT). The pie charts indicate the components of $NO_y$ at various segments along the ship's track (ONs = organic nitrates, pNit = particulate nitrate). $HNO_3$ was calculated via: $HNO_3 = NO_z - (ONs + pNit + NO_x + ClNO_2 + HONO)$.


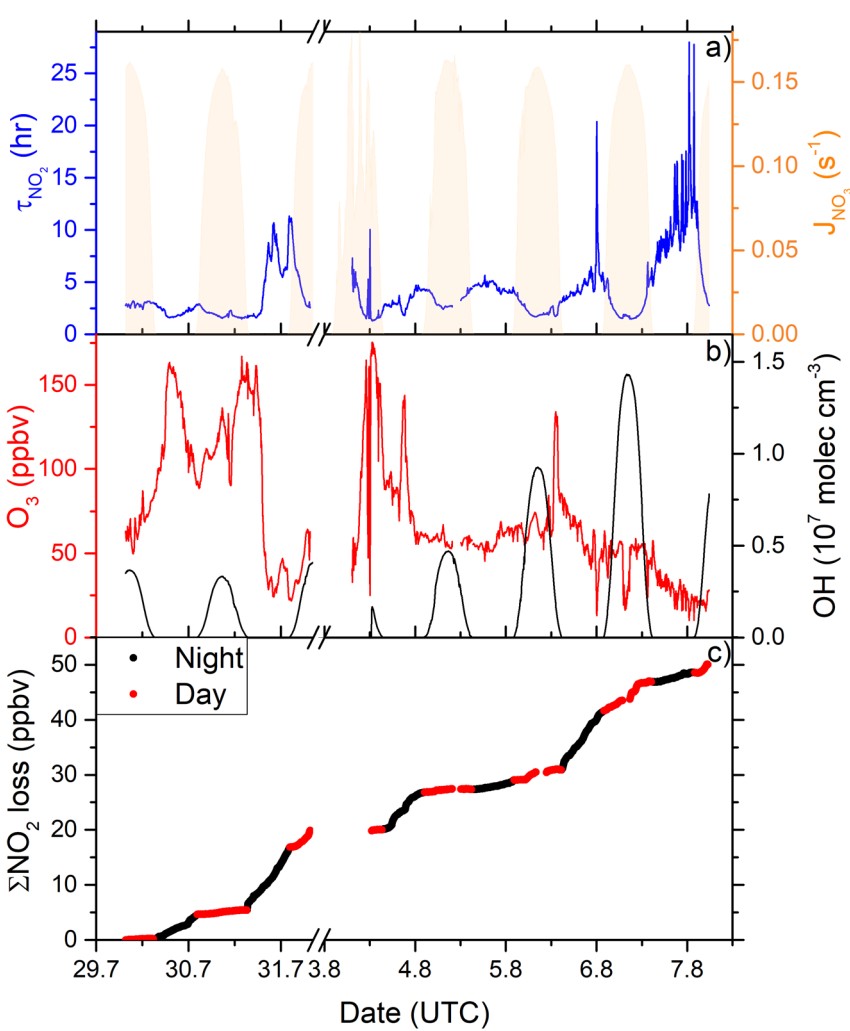

**Figure 12:** *(a)* Lifetime (τ) of NO$_2$ due to reactions with OH and O$_3$ in the *Arabian Gulf*, together with concentrations of O$_3$ and OH. The OH trace is an interpolation based on OH measurements and $J_{O1D}$ (see Sect. 3.3.3). Daytime hours are indicated via $J_{NO3}$. *(b)* Cumulative loss of NO$_2$ during the displayed time frame, based on the calculated lifetimes and measured NO$_2$.





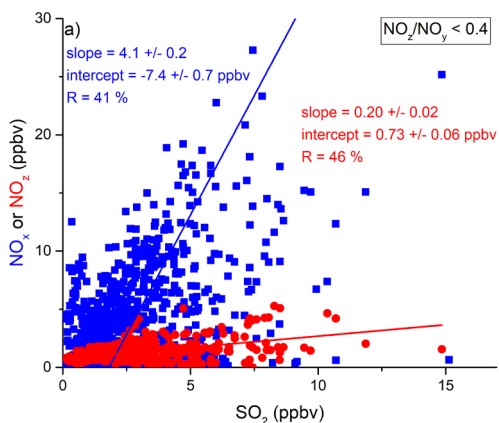

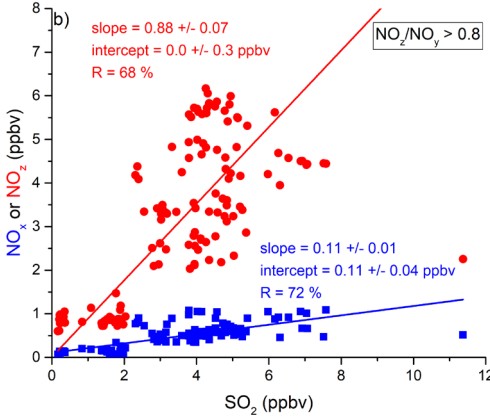

**Figure 13:** Correlation between $SO_2$ and $NO_x$ or $NO_z$ for *(a)* fresh and *(b)* aged $NO_x$ emissions in the *Arabian Gulf*.





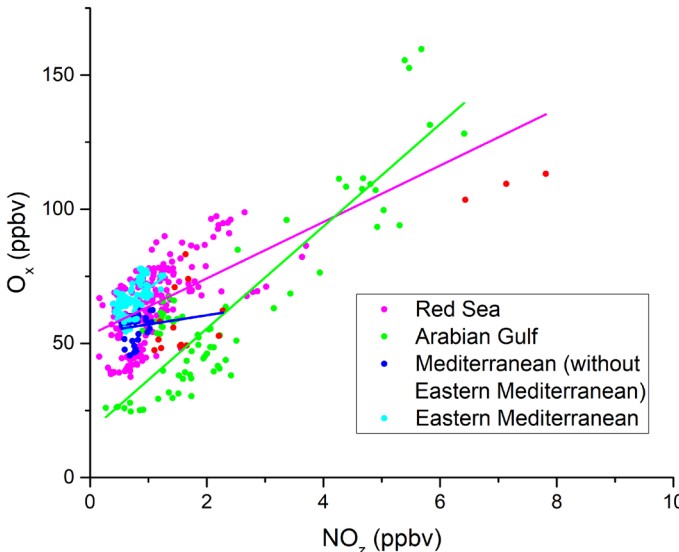

**Figure 14:** Correlation between $O_x$ (= $O_3$ + $NO_2$) and $NO_z$ during AQABA, with the regions indicated via the colour code. Only daytime measurements were used in this analysis. The OPEs for AQABA and for the individual regions shown in Table 3 were derived from linear fits of these data points. A clear regional variability can be observed for $O_x$ and $NO_z$ mixing ratios. Elevated $O_x$ and $NO_z$ levels were measured in the Arabian Gulf and the Red Sea.





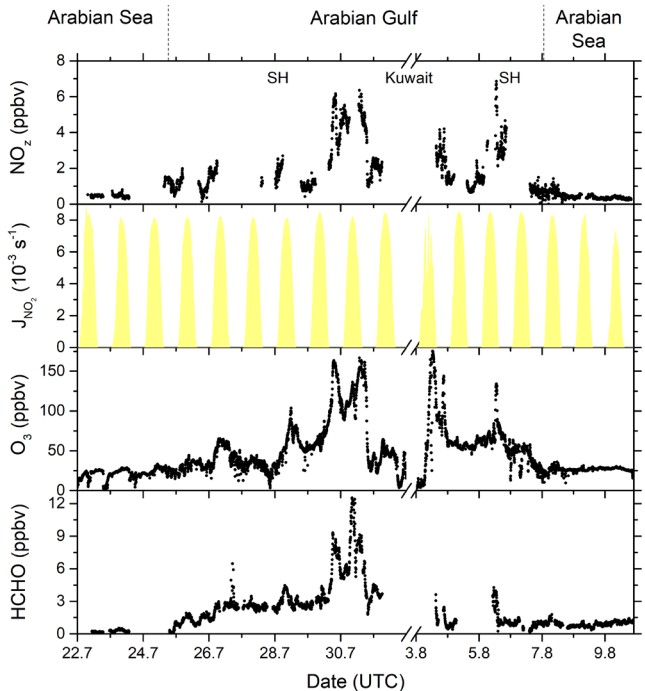

**Figure 15:** $NO_z$, $O_3$ and HCHO mixing ratios, together with $NO_2$ photolysis rates, during the transitions between the Arabian Sea and Gulf, and in the Arabian Gulf.


**Table 1:** Summary of correlation results between $NO_x$/$NO_z$ and $SO_2$ in all regions.

| Region | $NO_z$ / $NO_y$ | Species | Slope | Intercept (ppbv) | R (%) |
|---|---|---|---|---|---|
| *Mediterranean Sea* | < 0.4 | $NO_x$ | $4.0 \pm 0.1$ | $-1.9 \pm 0.3$ | 84 |
| | | $NO_z$ | $0.09 \pm 0.01$ | $0.69 \pm 0.03$ | 38 |
| | > 0.8 | $NO_x$ | $0.16 \pm 0.01$ | $0.049 \pm 0.005$ | 59 |
| | | $NO_z$ | $0.81 \pm 0.05$ | $0.39 \pm 0.02$ | 64 |
| *Red Sea* | < 0.4 | $NO_x$ | $3.7 \pm 0.1$ | $-2.2 \pm 0.3$ | 61 |
| | | $NO_z$ | $0.03 \pm 0.01$ | $0.87 \pm 0.03$ | 11 |
| | > 0.8 | $NO_x$ | $0.20 \pm 0.01$ | $0.07 \pm 0.01$ | 61 |
| | | $NO_z$ | $1.25 \pm 0.04$ | $0.40 \pm 0.03$ | 85 |
| *Arabian Gulf* | < 0.4 | $NO_x$ | $4.1 \pm 0.2$ | $-7.4 \pm 0.7$ | 41 |
| | | $NO_z$ | $0.20 \pm 0.02$ | $0.73 \pm 0.06$ | 46 |
| | > 0.8 | $NO_x$ | $0.11 \pm 0.01$ | $0.11 \pm 0.04$ | 72 |
| | | $NO_z$ | $0.88 \pm 0.07$ | $0.0 \pm 0.3$ | 68 |

**Table 2:** Average required production rates to maintain the observed $NO_x$ mixing ratios in aged air masses during the *Mediterranean Sea* transit and contributions from processes R19-R21.

| | Mediterranean Sea |
|---|---|
| $P_{chem} \pm SD$ ($10^5$ molec cm$^{-3}$ s$^{-1}$) | $2.8 \pm 2.2$ |
| $P(HONO+h\nu) \pm SD$ ($10^5$ molec cm$^{-3}$ s$^{-1}$) | $13.1 \pm 9.1$ |
| $P(pNit+h\nu) \pm SD$ ($10^5$ molec cm$^{-3}$ s$^{-1}$) | $1.8 \pm 0.4$ |
| $P(OH+HNO_3) \pm SD$ ($10^5$ molec cm$^{-3}$ s$^{-1}$) | $0.14 \pm 0.06$ |
| Number of data points (5 min averages) | 90 |

**Table 3:** Ozone production efficiencies (OPEs) for AQABA and the individual regions.

| | AQABA | Eastern Med. Sea | Red Sea | Arabian Gulf |
|---|---|---|---|---|
| OPE | $14.1 \pm 0.7$ | $15.4 \pm 2.4$ | $10.5 \pm 0.9$ | $19.1 \pm 1.1$ |
| Correlation Coeff. R (%) | 65 | 55 | 65 | 89 |
| $k_{NO_x}^{OH}$ / $k_{total}^{OH}$ (%) [a] | | 1.0 | 2.0 | 7.5 |
| $O_3$ (ppbv) [b] | | 58-73 | 42-81 | 23-108 |
| $NO_z$ (ppbv) [b] | | 0.5-1.0 | 0.5-2.1 | 0.9-4.9 |
| $NO_y$ / CO (%) [b],[c] | | | 1.4-7.0 | 1.9-14.6 |
| $OH_{max}$ ($10^6$ molec cm$^{-3}$) [d] | | 9.1 | 5.7 | 11.8 |

    *(a)   Median*
    *(b)   10 – 90 percentiles*
    *(c)   No CO data after 16 August 2017*
    (d)   *Average of daily OH peak concentrations, no data before 18 July 2017*

