# Peer review of "Reactive nitrogen around the Arabian Peninsula and in the Mediterranean Sea during the 2017 AQABA ship campaign."

_Atmospheric Chemistry and Physics, 2021_

## Author Comment (AC1)

**Author's Response to Referee #1**

*In this response, the referee comments (in black) are listed together with our replies (in blue) and the changes to the original manuscript (in red).*

**Summary**: The authors present reactive nitrogen measurements from the recent AQABA campaign, a ship-based summertime field campaign in the Mediterranean Sea, Red Sea, and the Arabian Gulf. Using a thermal dissociation cavity ring down spectrometer, they measure NOx (NO + NO2) and other reactive nitrogen species NOz (HNO3 + HONO + pNit + ClNO2 + N2O5 + others). Using these measurements and other auxiliary observations, they derive mean NO2 concentrations and lifetimes, cumulative NO2 losses during the day and night, and mean ozone production efficiencies. They also use back trajectories and ratios of NOz / NOy and NO2 / SO2 to attribute their observations to local ship emissions, near-coast industrial activities, and longer range transport.
This is a very well put together manuscript. The results are robust and thorough, and very useful for the community, especially given the lack of in situ data in the Red Sea and Arabian Gulf. I would recommend publication, following a few minor edits/suggestions for the authors first.

We thank the referee for the positive assessment of our paper and the helpful comments, which we address in the following responses.

**General comments:**

In the abstract (i.e. page 1 line 15) and in the methods section (i.e. page 5, line 18), the TD-CRDS instrument is presented as the primary instrument for both NOx and NOz, but it seems that the NOx measurements are all from the CLD instrument, which should be clarified. Also, were the NOz measurements derived from NOy – NOx(CRDS) or NOy – NOx(CLD)?

The lifetime and loss of $NO_x$ sections (i.e. 3.1.3, 3.2.3, and 3.3.3) rely on the CLD measurements. We have clarified this by adding a statement to the abstract, and by dedicating a separate methods sub-section to the CLD instrument. $NO_z$ mixing ratios, however, were always derived from $NO_y$ – $NO_x$ (TD-CRDS). The $NO_x$ vs. $SO_2$ correlations are also based on the TD-CRDS measurements.

*Abstract:* Complementing the TD-CRDS measurements, NO and $NO_2$ data sets from a chemiluminescence detector (CLD) were used in the analysis.

*Sect. 2.2:* NO and $NO_2$ were measured with a chemiluminescence detector (CLD 790 SR, ECO Physics, 5 s time resolution) as described in Tadic et al. (2020), with total measurement uncertainties of 6 % (NO) and 23 % ($NO_2$) and detection limits of 22 pptv for NO and 52 pptv for $NO_2$, both calculated at a time resolution of 5 s and a confidence interval of 2σ. The CLD detection method is based on the chemiluminescence of electronically excited $NO_2^*$ formed in the reaction of NO with $O_3$. Ambient $NO_2$ is photolytically converted to NO by exposure to UV light from LEDs emitting at wavelengths close to 398 nm. The CLD was calibrated very six hours using a 2 ppmv NO gas standard.

What was the methodology used to select the sub-regions in each Sea? Were they grouped together by eye? By some kind of filter? By similar HYSPLIT back trajectories? I ask because it looks like there is some data that is not considered in any of the sub-regions. If the sub-regions were selected by eye, I worry that it would bias the results in the paper to more "extreme" values, because that's what stands out most. Was there any attempt to do some kind of statistical analysis to group together regions with similar chemistry/emissions influence?

For the Mediterranean Sea, the sub-regions were selected based only on having homogeneous $NO_z/NO_y$ ratios. For the Red Sea and the Arabian Gulf, the definition of sub-regions was limited by the availability of TD-CRDS data, but also by other measurements of individual $NO_z$ components (e.g. gaps in ONs data set).
The overall goal was not to draw up a $NO_y$ budget for each point along the route, but to point out and contrast substantial changes in the chemical regimes in the different sub-regions. Concerning the sub-region definition, we added a short statement in Sect. 3.1.2:

The choice of these sub-regions was based on the existence of homogeneous $NO_z$ / $NO_y$ ratios over periods of hours to days. This approach enabled us to compare sub-regions with substantially different chemical regimes along the ship track, but does not lend itself to the derivation of a representative $NO_y$ budget for the entire region.

Related to the previous comment, do those sub-regions include an entire diurnal cycle? Do each of the sub-regions capture the same segment(s) of the diurnal cycle? If not, have the authors considered how this may change the relative contribution to NOz of the various species?

Not all sub-regions cover an entire diurnal cycle, the time frames vary between 4 (M1) and 46 hours (RS1) in length, with an average of 16 hours. In none of the major $NO_z$ components (pNit, $HNO_3$, ONs), were pronounced diurnal patterns observed. We therefore argue that any diurnal patterns in $NO_z$ and the $NO_z$ composition are overshadowed by the variability of air mass sources and the multi-component-nature of $NO_z$. We re-emphasize that the fractional contributions to $NO_y$ are to be considered as coarse estimates.

The data availaible in each sub-region did not always cover an entire diurnal cycle, which will have an impact on the fractional contributions of individual $NO_y$ species (see differences between day- and nighttime chemistry in Sect. 1). We argue however, that diurnal patterns in $NO_y$ are likely overshadowed by the variability of air mass sources. The $NO_y$ compositions presented are thus to be considered as coarse estimates.

**Specific comments:**

Page 4, Line 21: How long is the inlet upstream of the TD? Is there any concern about "sticky" gases such as HONO or HNO3 being trapped in the inlet walls before entering the TD?

The air samples reached the TD less than 30 cm behind the tip of the inlet.

Air samples reached the TD area less than 30 cm behind the tip of the inlet and we expect negligible inlet losses for $NO_y$ species.

Page 5, Line 14: The authors later explain that PM1 comes from the AMS instrument, but since that's in the next section, it's a bit unclear here whether the PM1 measurement is coming from the OPC.

We have now clarified that both $PM_{10}$ and $PM_1$ are derived from the OPC.

[…](both $PM_1$ and $PM_{10}$ were measured with the OPC).

Page 5, Line 33: Does the CIMS instrument output a HNO3 or a HONO measurement, even if uncalibrated? Can that signal be correlated with the calculated HNO3 or the LOPAP HONO measurement?

No, during AQABA the CIMS was not configured to monitor $HNO_3$ or HONO.

Page 6, Line 11: What is the path length of the HONO instrument? Is it co-located with the other instruments?

We added these pieces of information:

The path length of the instrument was 1.9 m, and the inlet was also located on the foredeck of the ship in ca. 5 m distance to the TD-CRDS inlets.

Page 6, Line 26: Were HYSPLIT back trajectories calculated at all locations along the ship track, or just those representative locations shown in the figures?

The back trajectories start at the geographical centers of the sub-regions. We also generated back trajectories starting at the ship's location 4 hours before/after passing these points to confirm that no changes in the general directions of the air mass origins occurred. For clarity of the graphs we chose to include only one trajectory per sub-region. We added in Sect. 3.1.2:

For visual clarity, only the back trajectories starting at the geographical centres of the respective sub-regions are displayed in Fig. 3. Back trajectories starting at the ship's location 4 hours before or after confirmed that the air-mass origin was very similar.

Page 9, Line 23: Here and in a few other places, regions are described as being "influenced by land-based" pollution. However, it seems that depending on how populated the land is, "land-based" can either mean free from fresh emissions, or heavily influenced by fresh emissions. Can the authors clarify, or perhaps use terminology that describes the regions in terms of their influence from fresh emissions, either ship- or land-based?

When using this or similar terms, we refer to pollution sources. The text now reads:

[…] are influenced by  fresh emissions from landbased sources […]

*Page 17, line 12:* […] the land-based emission sources of $NO_x$ from urban/ industrialised areas gain in importance […]

Page 10, Line 11: Figure 4c doesn't really show the averaged NOx lost per day and per night, just the cumulative. Can the authors add those numbers to the figure, perhaps under the legend, or remove the reference to that figure in this line?

Average day- and nighttime losses have been added to the legends of Figures 4c, 8c and 12c.

Page 17, Line 20: How do the authors define "background mixing ratios of NOx" here?

We have added a definition. In this and in previous sections we also identified background conditions by filtering out data points where the $NO_z$ / $NO_y$ ratio was below a certain threshold.

Background conditions refer to $NO_x$ mixing ratios found during periods when ship plumes were rarely encountered. "Background" $NO_2$ varied from region to region and was e.g. 50-150 pptv in the Mediterranean Sea).

Page 20, Line 5: How are OPE and NOPR defined differently here? It's a little unclear how these two parameters relate to each other.

We have added an explanation which distinguishes more clearly between the two metrics, at the place where we first compare OPE to NOPR results:

In contrast to the OPE, NOPR accounts for the total amount of $O_3$ produced in one day, considering production (governed by the formation of $NO_2$ via reactions of NO with $HO_2$ and $RO_2$) and loss (via photolysis and reaction with OH or $HO_2$). The OPE, on the other hand, focusses on the product side and assesses the competition between $O_3$ formation and sequestering into $NO_z$ from a given initial level of $NO_x$. By approximating the $O_3$ production rate via the $NO_2$ formation from NO reactions with $HO_2$ and $RO_2$, the NOPR thus neglects the alternative branch leading to $NO_z$.

Figure 14: The fitted light for the Eastern Mediterranean (light blue) is very difficult to see over the experimental data.

The colour has been changed to a darker tone.

**References**

Tadic, I., Crowley, J. N., Dienhart, D., Eger, P., Harder, H., Hottmann, B., Martinez, M., Parchatka, U., Paris, J. D., Pozzer, A., Rohloff, R., Schuladen, J., Shenolikar, J., Tauer, S., Lelieveld, J., and Fischer, H.: Net ozone production and its relationship to nitrogen oxides and volatile organic compounds in the marine boundary layer around the Arabian Peninsula, Atmos. Chem. Phys., 20, 6769-6787, 10.5194/acp-20-6769-2020, 2020.

---

## Author Comment (AC2)

**Author's Response to Referee #2**

*In this response, the referee comments (in black) are listed together with our replies (in blue) and the changes to the original manuscript (in red).*

Friedrich et al. report on measurements of 'reactive nitrogen' as part of the shipborne 2017 AQABA campaign. Data were acquired using a custom-built thermal dissociation cavity ring-down spectrometer recently described by Friedrich et al. (2020) supplemented by numerous auxiliary measurements. In this paper, the authors dive into some of the analysis of this rich data set, focussing on the conversion of $NO_x$ to $NO_z$ in 3 regions, on HONO budgets and on ozone production efficiencies.

The paper is written well. It is a bit too long, and there were some organizational shortcomings (see below) that should be addressed before the paper is accepted.

We thank the referee for the constructive review of our paper and the detailed comments, which we will address in the following responses.

**Major comments**

*1) Organizational / presentation issues*

$NO_z/NO_x$ ratios were investigated in 3 regions (3.1-3.3 - Mediterranean Sea, Red Sea, and Arabian Gulf). It is not sufficiently clear why the data set was divided in this way.

An extended explanation has been added in the introduction to Sect. 3.

Dividing the analysis into the three regions helps to highlight the chemically different environments encountered. An analysis of the Arabian Sea region was unfortunately not possible due to a gap in the $NO_z$ measurements between 9 and 17 August 2017, caused by instruement failure during heavy seas and winds. The division into the regions was based on the prevalent $NO_x$ mixing ratios displayed in Fig. S3c. In contrast to other AQABA publications (Eger et al., 2019; Pfannerstill et al., 2019; Tadic et al., 2020), the Gulf of Oman and the Suez Channel were included in the Arabian Gulf and the Red Sea regions, respectively, as a clear shift in $NO_x$ to mixing ratios below ca. 1 ppbv occurred both upon leaving the Gulf of Oman into the Arabian Sea and upon exiting the Suez Channel to the north towards the Mediterranean Sea. The transitions between the Arabian Gulf and the Gulf of Oman, and between the Northern Red Sea and the Suez region are less obviously represented in the $NO_x$ levels.

(b) A brief but incomplete analysis of HONO is presented in section 3.4. The introduction does not mention HONO (other than in R5) and this section appears "out of the blue". Consider removing this section if the plan is to write a separate paper on HONO anyways.

We would preferably keep Sect. 3.4 in the manuscript in its current form, introducing the potential role of HONO as a $NO_x$ source on AQABA, as this is an important finding of this study. A more comprehensive analysis of the HONO data set is outside the scope of this paper, but will be subject of a future publication. We thank the reviewer for pointing out that Sect. 3.4 is not sufficiently included into the flow of the main text, which has been amended in several places:

a) HONO photolysis as a source of $NO_x$ is now mentioned in the introduction:

$NO_x$ can be reformed from HONO at daytime through photolysis, with a noontime lifetime of ca. 20-30 minutes (Stutz et al., 2000).

b) Sect. 3.4 is now referenced in the outline part of the introduction:

In this paper we present $NO_x$, $NO_y$ and $NO_z$ mixing ratios obtained by a thermal dissociation cavity-ringdown-spectrometer (TD-CRDS), together with a comprehensive set of ancillary measurements and an analysis of the results in terms of photochemical processing/aging of air masses, chemical sources of $NO_x$ (e.g. from the photolysis of HONO), and the efficiency of ozone formation.

c) Sect. 3.4 is now referenced at the beginning of the results section (Sect. 3):

Chemical sources of $NO_x$ e.g. from the photolysis of HONO or pNit are discussed in Sect. 3.4.

An additional literature source has been added to strengthen the discussion on HONO linked to ship emissions.

Ship-derived HONO has a substantial effect on the rates of photochemical $O_3$ formation in the remote marine boundary layer, largely as a result of higher $RO_x$ production rates (Dai and Wang, 2021).

(d) Information already presented elsewhere should be removed and the previous paper(s) cited - e.g., Fig 1 of this paper is similar to Fig 1 / Fig 3 of Tadic et al. (2020), Fig 9 and 13 and Table 1 of this paper and Fig 7 of Eger et al. (2019). Likewise, rather than describing the TD-CRDS over 2 pages (section 2.1 - page 4 and 5) I would suggest simply citing the earlier Friedrich et al. (2020) manuscript.

Figure 1a has been moved to the supplement.

Section 2.1 has been substantially shortened to focus on information which is not given in Friedrich et al. (2020) and which is relevant to the presented AQABA data sets (see revised manuscript version with 'track changes').

(e) Pages 2-3. The section on nitrogen oxide chemistry in the introduction is written well, but in my opinion is not needed - similar sections of text have been presented

numerous times, including by the authors themselves in recent papers. It would have been more informative and interesting to tell the reader about what made the AQABA campaign interesting and worthwhile (e.g., effects of temperature/climate and mineral dust on nitrogen oxides, special/unique NOx sources in the regions etc.) and add more background on ozone production efficiencies (see g).

The chemistry part of the introduction has been shortened, focusing on reactions involving $NO_z$ species which are later discussed in the results section (see revised manuscript version with 'track changes').

The scientific motivation for the AQABA campaign has been extended:

Emissions from oil exploration provide a complex atmospheric mixture of $NO_x$ and anthropogenic VOCs. The presence of desert dust can have a significant impact on the budget of inorganic acids such as $HNO_3$. Finally, the overall elevated temperatures and actinic fluxes on AQABA promote rapid photochemical processing of $NO_x$. We therefore expect a more varied and complex chemistry than found in remote marine locations.

See g) for the novel paragraph on the OPE in the introduction.

(f) A table summarizing the various measurements and techniques would help.

Such a table is now provided (see Table 1).

(g) The OPE values calculated need to put more into context of existing literature (pg 19 lines 24-30). Consider stating in the introduction what values are typical or would be expected and expand the discussion.

The OPE explanation has been moved to the introduction and extended with more literature context:

The ozone production efficiency (OPE), a metric used in the analysis of the $O_3$ formation, quantifies the fractional transformation of primarily emitted $NO_x$ to $O_3$ (Liu et al., 1987; Trainer et al., 1993) and thus reflects the relative importance of competing photochemical processes leading to $O_3$ and $NO_z$ formation from $NO_x$. High values of OPE are favoured by low OH and VOC concentrations and values exceeding 80 have been reported for remote marine environments. Low, single digit values have been observed in polluted urban environments (Rickard et al., 2002; Wang et al., 2018). The location-dependence of the OPE can be further classified with previous observations from the literature. Minimal OPEs in urban environments between 1 and 2 have been reported from the Beijing area (Lin et al., 2011; Ge et al., 2013) and from the USA (Daum et al., 2000; Sillman, 2000; Nunnermacker et al., 2004). In rural and suburban environments, the OPE can increase to values between 10 and 15, as demonstrated in Northern America (Olszyna et al., 1994; Roussel et al., 1996; Fried et al., 1997; Ninneman et al., 2017) and in China (Sun et al., 2010). From oceanic samples, OPEs

of 65 and 87 were observed on the south-eastern coast of the UK (Rickard et al., 2002) and on Sable Island, Canada (Wang et al., 1996). Flights over the Western Pacific Ocean found values of 102-246 in the tropical area (latitude 0-18 °N), and of 73-209 further north (18-42 °N) (Davis et al., 1996). For the AQABA campaign, we expect lower OPEs than those observed in remote oceanic locations, due to the variable influx from harbours, coastal pollution and surrounding ship traffic.

(2) (Perceived) lack of novelty.

There have already been at least 7 papers presenting results from the AQABA campaign, yet the introduction avoids telling the reader what was presented in the earlier papers. In the introduction, it should make clear to the reader what new information and/or analysis are presented in this paper and why this paper is worthwhile. In particular, it should be stated how this paper differentiates itself from Tadic et al. (2020), "Net ozone production and its relationship to nitrogen oxides" to avoid the perception of duplication (in particular of section 3.5).

We have now included a summary of previous AQABA papers into the introduction and explain the novelty of this paper:

Previous analyses from this campaign focussed on sources and sinks of non-methane hydrocarbons (Bourtsoukidis et al., 2019), the role of OH reactivity in ozone chemistry (Pfannerstill et al., 2019), formation of $ClNO_2$ (Eger et al., 2019), ethane and propane emissions from the Red Sea (Bourtsoukidis et al., 2020), emission factors in ship plumes (Celik et al., 2020), marine emissions of methane sulfonamide (Edtbauer et al., 2020), rates of net $O_3$ production (Tadic et al., 2020), and the abundance of carbonyl compounds .
In this paper we present $NO_x$, $NO_y$ and $NO_z$ mixing ratios obtained by a thermal dissociation cavity-ringdown-spectrometer (TD-CRDS), together with NO and $NO_2$ mixing ratios from a chemiluminescence detector , a comprehensive set of ancillary measurements and an analysis of the results in terms of photochemical processing/aging of air masses, chemical sources of $NO_x$ (e.g. from the photolysis of HONO), and the efficiency of ozone formation..

Following a comment from referee #1 we explained the differences between OPE and NOPR in Sect. 3.5:

In contrast to the OPE, NOPR accounts for the total amount of $O_3$ produced in one day, considering production (governed by the formation of $NO_2$ via reactions of NO with $HO_2$ and $RO_2$) and loss (via photolysis and reaction with OH or $HO_2$). The OPE, on the other hand, focusses on the product side and assesses the competition between $O_3$ formation and sequestering into $NO_z$ from a given initial level of $NO_x$. By approximating the $O_3$ production rate via the $NO_2$ formation from NO reactions with $HO_2$ and $RO_2$, the NOPR thus neglects the alternative branch leading to $NO_z$.

(3) NOx lifetime - pg 9 line 30.

The equation given here is too simplistic in my opinion. Equation (1) should account for $N_2O_5$ formation, which can increase the $k_{13}[O_3]$ term by a factor of up to 2 (see Brown, S. S., et al. (2004), Nighttime removal of $NO_x$ in the summer marine boundary layer, *Geophys. Res. Lett.*, *31*(7), L07108, doi:10.1029/2004GL019412.). There are also sinks such as the heterogeneous conversion of $NO_2$ to $HONO/HNO_3$ that may need to be considered (mentioned on page 18, lines 8-).

We have added explanations for the omission of these two processes:

By using Eq. 1 to approximate the $NO_2$ loss rate constant, we neglect two further processes which can, under some conditions, influence the lifetime of $NO_2$. Our approach assumes that the nighttime formation of $NO_3$ leads to the removal of one $NO_2$ molecule. This approach would be invalid, if a significant fraction of $NO_3$ would be lost via formation (and subsequent heterogeneous loss) of $N_2O_5$. Firstly, we note that formation of $N_2O_5$ was hindered during AQABA by the high gas-phase reactivity of $NO_3$ towards VOCs (Eger et al., 2019) and that the transfer of $N_2O_5$ to the particle phase was hindered by high temperatures. For example, taking an $N_2O_5$ uptake coefficient $\gamma_{N2O5}$ of 0.03 (as found for polluted marine environments by Aldener et al. (2006)) and the median nighttime aerosol surface area (ASA) in the Mediterranean Sea of 1.78 x $10^{-6}$ $cm^2$ $cm^{-3}$ (Eger et al., 2019), we estimated a loss rate constant for uptake of $N_2O_5$ of 3.5 x $10^{-4}$ $s^{-1}$, which is two orders of magnitude lower than the rate constant (4.9 x $10^{-2}$ $s^{-1}$) for thermal decomposition at 25.7 °C (the mean, minimum nighttime temperature in the Mediterranean Sea).
We also neglect the loss of $NO_x$ via uptake of $NO_2$ onto black carbon (BC) particles. Using a literature uptake coefficient $\gamma_{NO2}$ of ca. 1 x $10^{-4}$ (Longfellow et al., 1999) and the aforementioned ASA, the first order loss rate constant for the heterogeneous uptake would be 1.8 x $10^{-6}$ $s^{-1}$. Using an $O_3$ mixing ratio of 63.4 ppbv (= nighttime median mixing ratio in the Mediterranean Sea), we calculate a first-order loss rate constant for the reaction of $NO_2$ and $O_3$ of 5.5 x $10^{-5}$ $s^{-1}$, which implies that > 95 % of total $NO_2$ loss at nightime $NO_2$ is due to $O_3$. Uptake of $NO_2$ might therefore be relevant for HONO formation (see Sect. 3.4), but does not constitute a relevant loss process for $NO_x$.

**Minor/Specific comments**

In the future, please number continuously and do not restart numbering on each page. You are creating more work for the reviewer which is in nobody's interest.

Thank you for the advice, which we will be following in future publications. We had been using the .docx template provided by Copernicus without further considering the line numbering.

Abstract, line 20 - HONO. Consider stating how the role of HONO was assessed (were there measurements?)

Has been added:

The role of HONO was assessed by calculating the $NO_x$ production rate from its photolysis.

Abstract line 24 - OPE. Consider stating how OPE were calculated (plots of $\Delta O_3$ vs $\Delta NO_z$ ?)

We now mention the method in the abstract:

Regional ozone production efficiencies (OPE; calculated from the correlation between $O_x$ and $NO_z$, where $O_x = O_3 + NO_2$) ranged from […]

pg 1 line 8 - Please define "M"

[…] where M is a collision partner.

pg 3 line 2 - "lifetimes of a few hours". The lifetime of PAN may be much longer aloft are in Arctic environments.

The sentence has been removed while shortening the introductionas requested.

pg 4 line 19 - please specify the make/model of the 3-way valve and state what the internal surfaces are made of.

The valve is made out of PTFE and was obtained from Neptune Research, Inc. (type 648T032, orifice diameter 4 mm). This information, however, was already given in Friedrich et al. (2020) and was removed from the revised manuscript in order to shorten the paper as requested by the referee.

pg 4 line 31 "adding 19 ppmv of O3" - Please clarify if this mixing ratio refers to the amount of O3 (in O2?) added (in which case also state the flow) or if "19 ppmv" refers to the amount of O3 after addition to the sampled air.

The $O_3$ mixing ratio of 19 ppmv was detected in the sampling flow of the TD-CRDS, i.e. after diluting in ca. 3 slm synthetic air. This information, however, was already given in Friedrich et al. (2020) and was removed from the revised manuscript in order to the paper as requested by the referee.

pg 5 line 4. Does the TD-CRDS respond to nitrate associated with mineral dust which may occur in the study area (e.g., https://acp.copernicus.org/articles/16/1491/2016/)?

We expect only a weak (if any) response to nitrates on mineral dust, as we write in Sect. 3.1.2:

"Detection of coarse mode pNit by the TD-CRDS (see Friedrich et al. (2020)) would lead to an overestimation of $HNO_3$. However, given that the thermal dissociation to $NO_2$ of $NaNO_3$ particles with 300 nm diameter is inefficient (~ 20 %) with this instrument, a significant bias by coarse mode nitrate (e.g. associated with sea salt or mineral dust) appears unlikely."

In Sect. 2.1 we also discuss that coarse mode nitrates were only encountered in short periods in high abundancies:

"The fractional contribution of coarse-mode particles to the overall mass concentration were derived using data from an Optical Particle Counter (OPC) and via the ($PM_{10}$-$PM_1$)/$PM_{10}$ ratio (both $PM_1$ and $PM_{10}$ were measured with the OPC). We see from Fig. S1 that the impact of coarse mode nitrate may have been largest on both legs in the transitional area between Southern Red Sea and Arabian Sea, where OPC $PM_{10}$ mass concentrations exceeded 150 µg $m^{-3}$ and the coarse mode fraction was consistently > ca. 90 %."

pg 5 line 20. Please state here how the detection limit was defined (move up from line 27) and also state how long data were averaged (longer averaging times => better detection limits).

The information has been moved and refined:

The total uncertainty (at 50 % relative humidity and one minute integration time) amounts to 11 % + 10 pptv for $NO_x$ and to 16 % + 14 pptv for $NO_z$ if we disregard the non-quantitative detection of coarse-mode, non-refractory nitrate (see below). Detection limits (5 s integration time) during the AQABA campaign were 98 pptv for $NO_x$, 51 pptv for $NO_y$, and 110 pptv for $NO_z$ and are higher than those reported for laboratory operation owing to problems with optical alignment due to the motion of the ship. Detection limits are defined as the 2σ standard deviation between consecutive zeroing periods. Under laboratory conditions, $NO_x$ detection limits of 40 pptv (1 min average) were obtained (Friedrich et al., 2020); 6 pptv (40 s) have been achieved with undegraded mirrors (Thieser et al., 2016).

pg 5 line 26. Why correct for humidity?

The discussion of systematic errors has been removed due to overlap with Thieser et al. (2016) and Friedrich et al. (2020). Correcting for ambient humidity is necessary due to the Rayleigh scattering of water and the zeroing of the TD-CRDS with dry synthetic air.

pg 6 line 14-16 How can the uncertainty of j data be 10% if upwelling radiation was not included?

We now write:

The overall uncertainty in *J* is ca. 15 %, which includes calibration accuracy (Bohn et al., 2008) and the negelect of upwelling radiation from the sea-surface

pg 6 line 25. Which meteorological field was used for the HYSPLIT trajectories?

[…], using the Global Data Assimilation System (GDAS1) meteorological model.

pg 17 section 3.4 "HONO formation"

The analysis appears to be only considering daytime processes in this section. How does the HONO budget during AQABA compared to the observations by Wojtal et al. (2011)?

We have included a remark about the daytime limitation of the analysis and give reference to the possibility of a nighttime pseudo stationary state between gas-phase HONO and HONO adsorbed to marine surfaces.

We emphasise that the analysis presented here focussed on the daytime chemistry of HONO. At nighttime, a pseudo stationary state, independent of fresh $NO_x$ input, has been observed by Wojtal et al. (2011), and explained with a reversible deposition of HONO on marine surfaces. This will however be insignificant during the day.

pg 20 line 30. "HONO photolysis was as a significant source of NOx." It is not a net source if HONO is generated from NOx.

This comment has misunderstood the point we are making:

We argue that HONO is not generated photochemically from $NO_x$ (i.e. from NO + OH) but from emissions of ships. If ships exhaust contains HONO (e.g on particles) then the $NO_x$ that comes from HONO is not simply recycling $NO_x$.

**References**

[revised manuscript text omitted]

---

## Author Response (AR2)

**Author's Response to Editor Corrections**

Technical Corrections (line numbers refer to track changes version)

1) Line 54: Incomplete sentence. Please revise

2) Line 54-55: "Reactions with organic peroxy radicals converts both NO and $NO_2$ to organic nitrates or peroxy nitrates..." This wording is confusing. Please consider something along the lines of "Reactions of organic peroxy radicals converts $NO_2$ to peroxy nitrates and NO to organic nitrates (minor channel) or $NO_2$ (major channel)."

3) Line 75: Given the discussion of $NO_3$ + VOC reactions later in the manuscript, it is worth mentioning that reaction pathway for $NO_3$ here as well.

4) Figure 3: The first time through, it took me a bit to find the legend for the pie charts. I suggest including the colors in the figure caption and increasing the size of the labels in the figure. Since subsequent figures of the same type (e.g., Fig. 7, etc.) use the same color scheme, I think it is ok to leave the subsequent figures as is.

5) Line 322-321: The switch between discussing the composition of $NO_z$ and $NO_y$ mid-sentence is confusing. I suggest splitting into 2 sentences or rephrasing so that only $NO_y$ or $NO_z$ is discussed.

All technical corrections have been implemented according to the Editor's comments. Please refer to the attached marked-up version of the manuscript.

[revised manuscript text omitted]